# Suppression of ischemia in arterial occlusive disease by JNK-promoted native collateral artery development

**Kasmir Ramo[1], Koichi Sugamura[2,3], Siobhan Craige[2,3], John F Keaney Jr[2,3], Roger J Davis[1,4]\***

[1]Program in Molecular Medicine, University of Massachusetts Medical School, Worcester, United States; [2]Cardiovascular Medicine Division, University of Massachusetts Medical School, Worcester, United States; [3]Department of Medicine, University of Massachusetts Medical School, Worcester, United States; [4]Howard Hughes Medical Institute, Worcester, United States

**Abstract** Arterial occlusive diseases are major causes of morbidity and mortality. Blood flow to the affected tissue must be restored quickly if viability and function are to be preserved. We report that disruption of the mixed-lineage protein kinase (MLK) - cJun NH$_2$-terminal kinase (JNK) signaling pathway in endothelial cells causes severe blockade of blood flow and failure to recover in the murine femoral artery ligation model of hindlimb ischemia. We show that the MLK-JNK pathway is required for the formation of native collateral arteries that can restore circulation following arterial occlusion. Disruption of the MLK-JNK pathway causes decreased Dll4/Notch signaling, excessive sprouting angiogenesis, and defects in developmental vascular morphogenesis. Our analysis demonstrates that the MLK-JNK signaling pathway is a key regulatory mechanism that protects against ischemia in arterial occlusive disease.

**\*For correspondence:** roger. davis@umassmed.edu

## Introduction

Ischemic stroke, myocardial infarction and peripheral artery disease result from arterial occlusion that blocks blood flow leading to severe tissue ischemia and necrosis. To prevent loss of tissue viability and function, blood flow to the affected tissue must be restored quickly. Collaterals are artery-to-artery or arteriole-to-arteriole interconnections that can bypass an occlusion by providing an alternative route for blood flow to the affected tissue that restores tissue homeostasis and limits tissue damage (*Antoniucci et al., 2002*; *Schaper, 2009*; *Faber et al., 2014*; *Simons and Eichmann, 2015*). Indeed, clinical outcome in patients with arterial occlusion depends on the presence of an adequate collateral circulation and animal models of arterial occlusion provide strong evidence for the critical importance of the extent of the native (pre-existing) collateral circulation in restoring blood perfusion and limiting ischemic sequelae following arterial occlusion.

Adequate restoration of blood flow depends on collateral artery size, number, and the pattern of connectivity, but also on functional adaptation to changes in blood flow. Following arterial occlusion, more blood flow is diverted to the collateral circulation and this increased flow and sheer stress in collateral arteries initiates a number of processes that result in the outward remodeling (arteriogenesis) of these vessels into efficient conductance arteries (*Heil et al., 2006*; *Schaper, 2009*; *van Royen et al., 2009*; *Simons and Eichmann, 2015*). Collateral artery remodeling involves multiple cellular processes, including endothelial cell activation and proliferation, monocyte/macrophage recruitment and smooth muscle cell proliferation, all of which contribute to increased collateral artery diameter,

**eLife digest** A blocked artery can have serious health consequences. For example, heart attacks and strokes are caused by such blockages. Artery blockages are harmful because tissues and organs are supplied with oxygen carried in blood cells and without adequate oxygen they may suffer damage or even die. The body has back up arteries or collateral arteries that help to mitigate the damage caused by a blood flow blockage. These arteries normally do not deliver blood to tissues. However, if an artery becomes blocked, these arteries can provide new and efficient routes of blood flow that can bypass the blockage.

How well patients fair after an artery blockage depends on them having a working system of collateral arteries. These collateral vessels must quickly spring into action and even widen to accommodate adequate blood flow. However, little is known about how the collateral arteries are formed.

Previous studies have suggested that an enzyme called cJun NH2-terminal kinase (or JNK for short) was essential for the formation of new blood vessels. Now, Ramo et al. show that while JNK is essential for the formation of collateral arteries during development, it is not required for the formation of blood vessels in adults. In the experiments, mice were genetically engineered to lack genes encoding two types of JNK in the cells that line the blood vessels. When these mice became adults they were still able to produce new blood vessels, just like normal mice. But the genetically engineered mice were missing the healthy collateral arteries that mice normally have in their muscles.

Next, Ramo et al. blocked arteries in the muscles of mice and found that mice without JNK suffered worse injuries than normal mice. Further experiments showed that JNK helps to regulate genes that control blood vessel formation during early development. Without JNK, developing blood vessels grow into poorly organized networks instead of forming normal collateral arteries. The experiments suggest that humans who have mutations in the genes for JNK or related genes may be more susceptible to the harmful effects of artery blockages in the muscles. More studies are now needed to test this hypothesis.

including increased thickness of the tunica media. These structural and functional adaptations depend on the presence of the native collateral circulation.

Although the collateral circulation is crucially important for the protective response to arterial occlusive diseases, little is known about the cellular and morphogenetic processes, or the molecular factors and mechanisms, that contribute to native collateral artery formation (*Antoniucci et al., 2002*; *Schaper, 2009*; *Faber et al., 2014*; *Simons and Eichmann, 2015*). However, studies of lepto-menengial (or pial) collateral arteries in the brain have provided significant insight. Murine pial collaterals are established during embryonic development with some remodeling and maturation continuing postnatally. The process of native collateral artery formation during embryogenesis has been termed collaterogenesis and involves a number of molecules including, platelet-endothelial cell adhesion molecule 1 (PECAM1) (*Chen et al., 2010*), gap junction protein, connexin37 (Cx37) (*Fang et al., 2011*, *2012*), prolyl hydroxylase domain-containing protein 2 (PHD2) (*Takeda et al., 2011*), endothelial nitric oxide synthase (eNOS) (*Dai and Faber, 2010*), chloride intracellular channel 4 (CLIC4) (*Chalothorn et al., 2009*), and Synectin (*Moraes et al., 2013*). Signaling pathways that have been reported to contribute to collaterogenesis include NF-κB (*Tirziu et al., 2012*), VEGF (*Chalothorn et al., 2007*; *Lucitti et al., 2012*), and the Dll4 – Notch pathway (*Cristofaro et al., 2013*).

The purpose of this study was to examine the role of the c-Jun NH$_2$-terminal kinase (JNK) (*Davis, 2000*), a signaling pathway that has been reported to play major roles in angiogenic responses (*Jiménez et al., 2001*; *Ennis et al., 2005*; *Medhora et al., 2008*; *Uchida et al., 2008*; *Guma et al., 2009*; *Shen et al., 2010*; *Kaikai et al., 2011*; *Ma et al., 2012*; *Du et al., 2013*; *Salvucci et al., 2015*). Our approach was to study the effect of compound gene disruption in endothelial cells to prevent JNK signaling in mice. We did not find that JNK signaling was required for

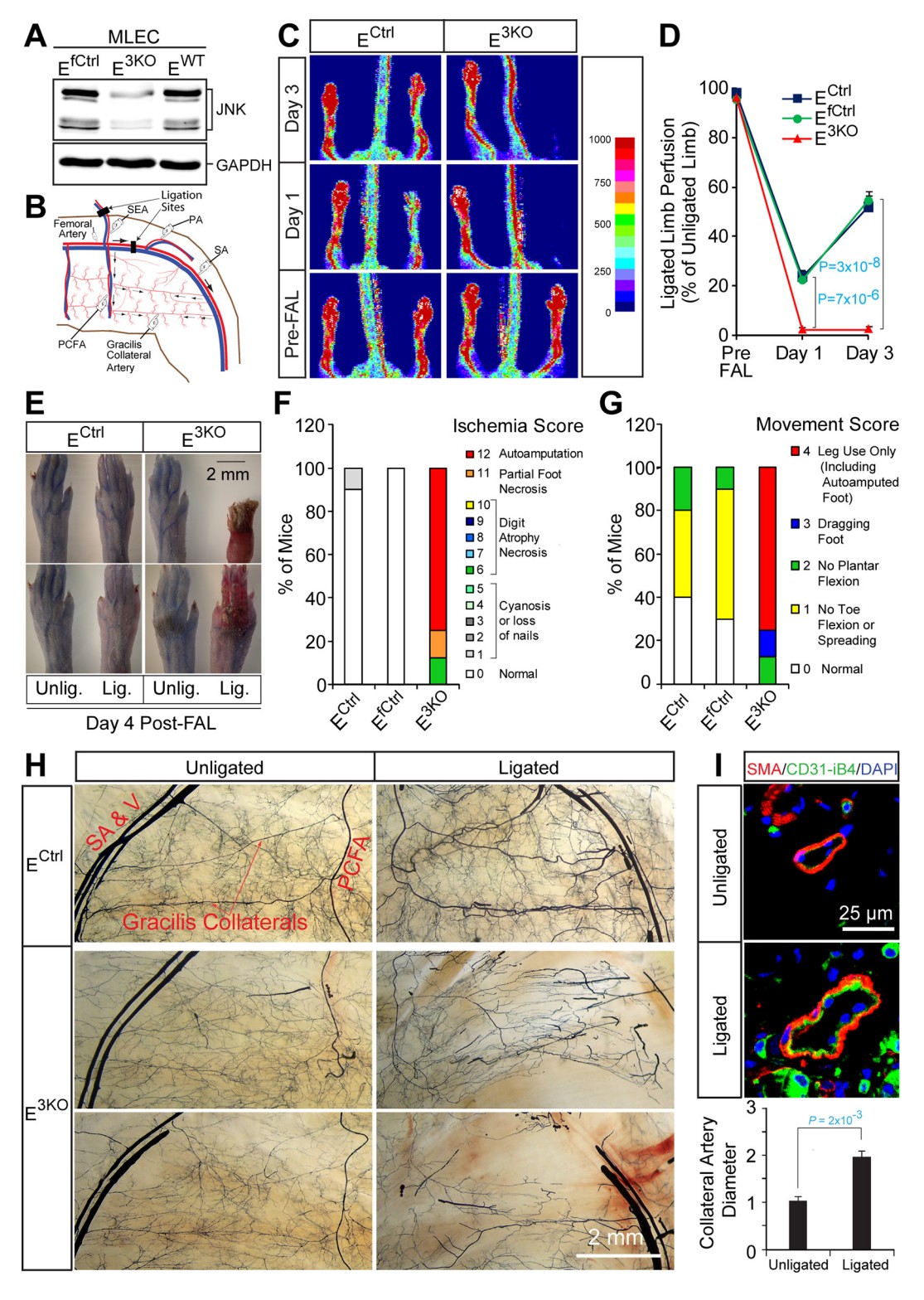

**Figure 1.** Enhanced blood perfusion blockade and severe ischemic injury in endothelial JNK-deficient mice upon arterial occlusion. (A) Control and JNK-deficient primary endothelial cells were examined by immunoblot analysis by probing with antibodies to JNK and GAPDH. (B) Simplified diagram of the medial aspect of the mouse hindlimb skeletal muscle vasculature. The common femoral artery (FA) and its main branches (proximal caudal femoral artery [PCFA], popliteal artery [PA] and saphenous artery [SA]) supply blood to the proximal and distal hindlimb. Ligation of the FA plus the superficial epigastric artery (SEA) as indicated, leads to reduced blood flow to the distal hindlimb, while flow through the PCFA and gracilis collaterals is

*Figure 1 continued on next page*

*Figure 1 continued*

enhanced. (C) Representative laser Doppler images showing blood perfusion (high perfusion red, no perfusion dark blue) in the hindlimbs of control and JNK-deficient mice prior to unilateral FA ligation (Pre-FAL) and on day 1 and 3 post-FAL. (D) Quantitation of hindlimb blood flow shows significantly enhanced blood perfusion blockade following FAL and no recovery 3 days after ligation in JNK-deficient mice compared to control mice (mean ± SEM; n = 7~10). (E) Representative images of mouse paws on Day 4 post-FAL. Lig., ligated; Unlig., contralateral unligated. (F,G) Quantitation of ischemic (F) and movement (G) scores for mice on Day 4 post-FAL (7~10 mice per group). (H) Representative whole mount preparations of the medial surface of Microfil-filled adductor muscle vasculature from day 4 post-FAL hindlimbs and contralateral unligated limbs. (I) The gracilis collateral arteries were stained for SMA and CD31/iB4. The artery diameter was quantitated (mean ± SEM; n = 10~12). Source data are included as *Figure 1—source data 1*.

The following source data and figure supplements are available for figure 1:

**Source data 1.** Source data for *Figure 1*.

**Figure supplement 1.** Characterization of endothelial JNK-deficient mice and lung endothelial cells.

**Figure supplement 1—source data 1.** Source data for *Figure 1—figure supplement 1*.

**Figure supplement 2.** Endothelial JNK-deficient mice have no major perturbations in the hematopoietic system.

**Figure supplement 2—source data 1.** Source data for *Figure 1—figure supplement 2.*

**Figure supplement 3.** Normal hypoxia responses and VEGF signaling in JNK-deficient endothelial cells.

**Figure supplement 3—source data 1.** Source data for *Figure 1—figure supplement 3*.

**Figure supplement 4.** Endothelial JNK is not required for proliferation, migration, and angiogenic responses in vitro.

**Figure supplement 4—source data 1.** Source data for *Figure 1—figure supplement 4*.

**Figure supplement 5.** Endothelial JNK is not required for in vivo pathologic angiogenesis.

**Figure supplement 5—source data 1.** Source data for *Figure 1—figure supplement 5*.

**Figure supplement 6.** Compound JNK-deficiency in endothelial cells causes defects in the response to arterial occlusion.

**Figure supplement 6—source data 1.** Source data for *Figure 1—figure supplement 6*.

**Figure supplement 7.** JNK deficient mice show no perturbations in overall cardiovascular function.

**Figure supplement 7—source data 1.** Source data for *Figure 1—figure supplement 7*.

**Figure supplement 8.** JNK-deficiency causes defects in artery size and connectivity, but not the hypoxia response after femoral artery ligation.

**Figure supplement 8—source data 1.** Source data for *Figure 1—figure supplement 8*.

angiogenesis in vitro or in adult mice, but JNK signaling is required for proper vascular morphogenesis and the normal formation of collateral arteries in muscle.

## Results

We tested the role of the JNK signaling pathway in endothelial cells using a conditional gene ablation strategy. In addition to the ubiquitously expressed JNK1 and JNK2 isoforms, JNK3 may also be expressed by some endothelial cells (*Pi et al., 2009*). We therefore established mice with endothelial deficiency of JNK1/2 ($E^{2KO}$) or all three JNK isoforms ($E^{3KO}$). Control mice included *Cre*[+] ($E^{WT}$ and $E^{Ctrl}$) mice and *Cre*[−] littermates ($E^{fCtrl}$). The $E^{2KO}$ and $E^{3KO}$ mice used in this study developed normally

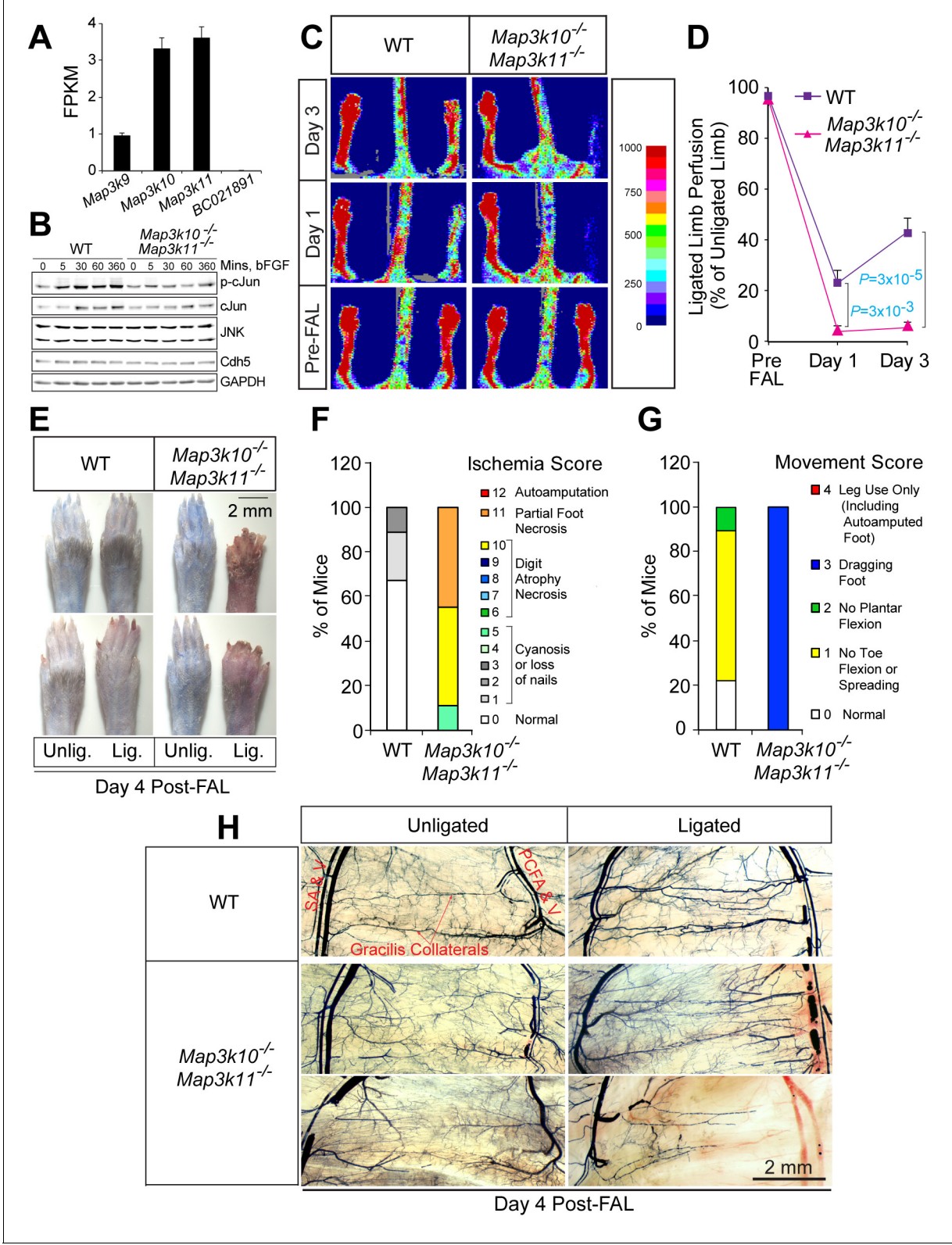

**Figure 2.** Severe ischemic injury in MLK2/3-deficient mice after femoral artery ligation. (**A**) The expression of members of the MLK protein kinase family (MLK1, MLK2, MLK3 & MLK4) in primary MLEC cultures was examined by measurement of *Map3k9, Map3k10, Map3k11*, and *BC021891* mRNA by RNA-seq analysis (mean fragments per kilobase of transcript per million mapped reads (FPKM) ± SEM; n = 6). (**B**) Primary wild-type (WT) and *Map3k10[-/-] Map3k11[-/-]* MLEC cultures were treated without and with 100 ng/ml bFGF and examined by immunoblot analysis by probing with antibodies to pSer[63]-
*Figure 2 continued on next page*

*Figure 2 continued*

cJun, cJun, JNK, Cdh5, and GAPDH. (**C**) Representative laser doppler images showing blood perfusion (high perfusion red, no perfusion dark blue) in the hindlimbs of WT and *Map3k10*$^{-/-}$ *Map3k11*$^{-/-}$ mice mice prior to unilateral FA ligation (Pre-FAL) and post-FAL. (**D**) Quantitation of hindlimb blood flow demonstrated that MLK2/3-deficient mice exhibited significantly increased blood perfusion blockade and no recovery by day 3 post-FAL compared with control mice (mean ± SEM; n = 7). (**E**) Representative images of mouse paws on Day 4 post-FAL. Lig., ligated; Unlig., contralateral unligated. (**F,G**) Quantitation of ischemic (**F**) and movement (**G**) scores for mice on Day 4 post-FAL (n = 9). (**H**) Representative whole mount preparations of the medial surface of Microfil-filled adductor muscle vasculature isolated from day 4 post-FAL hindlimbs and contralateral unligated limbs. Source data are included as *Figure 2—source data 1*.

The following source data is available for figure 2:

**Source data 1.** Source data for *Figure 2*.

and were healthy and fertile. We found no differences in body weight at birth and postnatal (P) day 6, but adult E$^{2KO}$ and E$^{3KO}$ mice were slightly smaller than control mice (*Figure 1—figure supplement 1A–C*). Primary murine lung endothelial cells (MLEC) isolated from E$^{3KO}$ mice (*Figure 1—figure supplement 1D,E*) demonstrated reduced JNK expression compared with MLEC from control mice (*Figure 1A*). Control studies demonstrated that JNK expression by hematopoietic cells, the number of circulating blood cells, and bone marrow function in transplantation assays were similar in control and E$^{3KO}$ mice (*Figure 1—figure supplement 2*).

## Angiogenesis does not require JNK signaling

We examined angiogenic responses of control and E$^{3KO}$ MLEC in vitro. JNK was not activated by hypoxia or VEGF and both control and E$^{3KO}$ MLEC mounted similar responses to hypoxia and VEGF (*Figure 1—figure supplement 3*). Tubulogenesis assays in matrigel demonstrated no differences between control and E$^{3KO}$ MLEC and no differences between control and E$^{3KO}$ mice were detected in VEGF-induced microvessel sprouting from collagen-embedded aortic rings (*Figure 1—figure supplement 4A,B*). We also found no differences in proliferation or migration between control and E$^{3KO}$ MLEC (*Figure 1—figure supplement 4C–E*). To assess angiogenesis in vivo, we examined laser-induced injury of the eye; no differences in choroidal neovascularization between control and E$^{3KO}$ mice were observed (*Figure 1—figure supplement 5A,B*). Similarly, we found no differences in tumor angiogenesis between control and E$^{3KO}$ mice (*Figure 1—figure supplement 5C–F*). Collectively, these data demonstrate that JNK in endothelial cells is not required for angiogenesis in vitro or in vivo in adult mice.

## JNK is required for preventing ischemia following arterial occlusion

To test the role of endothelial JNK in the response to arterial occlusion, we performed unilateral femoral artery ligation (FAL) on control and E$^{3KO}$ mice. This procedure causes hypoxia in the calf muscles that stimulates angiogenesis, but the proximal adductor muscles experience little or no hypoxia because of blood flow redistribution by the native collateral circulation (*Deindl et al., 2001*) (*Figure 1—figure supplement 1B*). We ligated the femoral artery (FA) between the proximal caudal femoral artery (PCFA) and the popliteal artery (PA) (*Kochi et al., 2013*); this is a mild version of the FAL procedure (*Figure 1B*). Laser Doppler imaging revealed ~80% decreased blood perfusion to the ligated limbs of control mice; blood perfusion was restored to ~60% of the contralateral limbs by day 3 (*Figure 1C,D*). These mice did not exhibit major hallmarks of ischemia (*Figure 1E–G*). In contrast, E$^{3KO}$ mice showed complete blockade of blood flow following occlusion (*Figure 1C,D*) leading to severe necrosis (*Figure 1E–G*). Ligation of the FA more proximally at its origin (a more severe form of FAL) also demonstrated increased blood flow blockade and failure to recover in both E$^{2KO}$ and E$^{3KO}$ mice (*Figure 1—figure supplement 6A–C*). In contrast, no post-FAL phenotype was detected in mice with JNK1 or JNK2-deficiency alone (*Figure 1—figure supplement 6D*) or in mice with JNK1 plus JNK2-deficiency in hematopoietic cells or muscle (*Figure 1—figure supplement 6G–I*). These data suggest that JNK1/2 in endothelial cells play a key role in the response to arterial occlusion. Consistent with this conclusion, coronary artery occlusion caused significantly greater mortality of E$^{3KO}$ mice compared with control mice (*Figure 1—figure supplement 6E,F*).

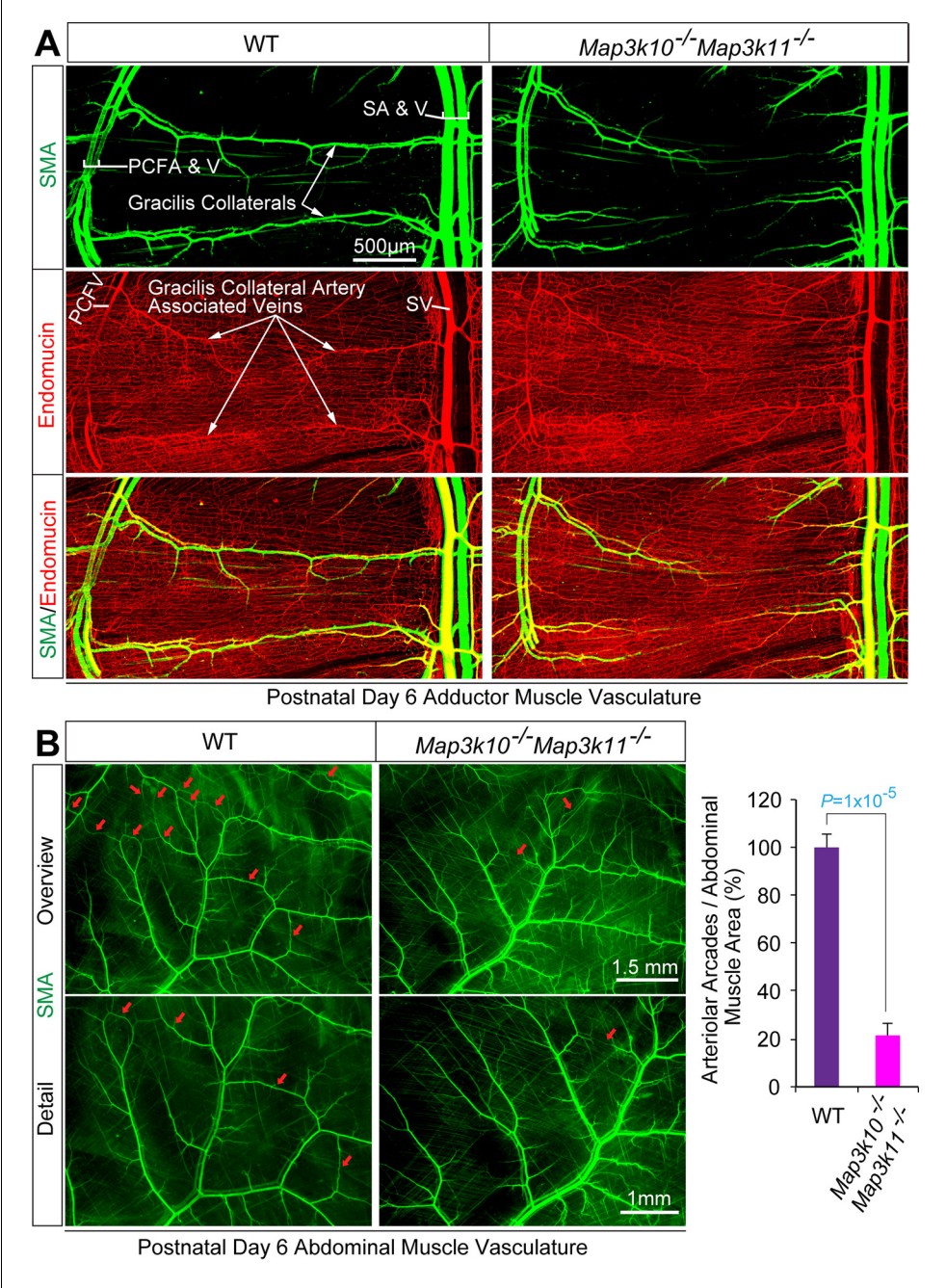

**Figure 3.** MLK2/3-deficient mice exhibit defects in native collateral artery formation. (**A**) Representative confocal images (n = 5 mice) of control and MLK2/3-deficient whole mount P6 adductor muscle vasculature stained with antibodies to endomucin (capillary and venous vasculature, red) and SMA (arterial and venous smooth muscle, green). Gracilis collateral arteries in WT mice, but not MLK2/3-deficient mice, interconnect the PCFA to the SA. (**B**) Representative stereomicroscope images of P6 whole mount abdominal muscle stained with an antibody to SMA (green). Arteriole-to-arteriole arcades are indicated (red arrows). The abdominal muscle vasculature of *Map3k10$^{-/-}$Map3k11$^{-/-}$* mice shows very few arteriole-to-arteriole interconnections. Quantitation reveals significantly reduced arteriolar arcade numbers in *Map3k10$^{-/-}$Map3k11$^{-/-}$* mice compared to WT mice (mean ± SEM; n = 5 mice). Source data are included as *Figure 3—source data 1*.

The following source data is available for figure 3:

**Source data 1.** Source data for *Figure 3*.

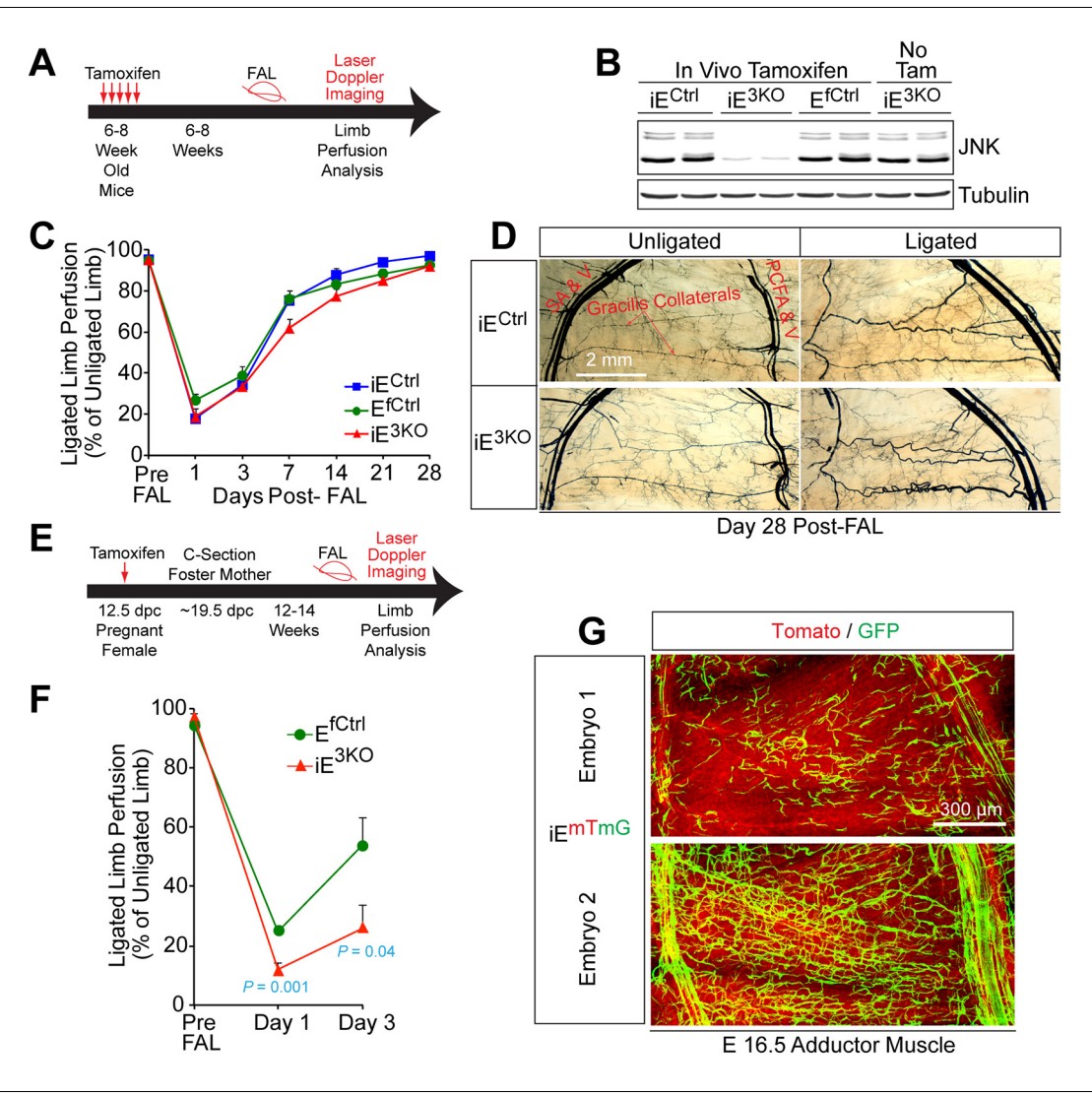

**Figure 4.** Endothelial JNK is not required for the arteriogenic response of gracilis collaterals in adult mice. (**A**) Timeline of tamoxifen administration to induce disruption of *Mapk8^{LoxP}* and *Mapk9^{LoxP}* alleles in the vascular endothelium of adult mice prior to FAL and analysis of blood flow by laser doppler imaging. (**B**) Primary MLEC cultures prepared from mice treated without and with tamoxifen were examined by immunoblot analysis by probing with antibodies to JNK and αTubulin. The data are representative of two independent MLEC isolations per group (2~3 mice used per cell preparation). (**C**) Quantitation of laser doppler analysis of limb blood flow demonstrated no significant differences (p>0.05) in blood perfusion blockade and recovery over 28 days post-FAL between tamoxifen-treated endothelial JNK-deficient mice and tamoxifen-treated control mice (mean ± SEM; n = 5~10). (**D**) Microfil perfusion of adductor muscle vasculature demonstrated the presence of similar gracilis collateral arteries in JNK-deficient and control mice and similar collateral artery remodeling at day 28 post-FAL. The images are representative of 5~8 mice per group. (**E**) Timeline of tamoxifen administration to induce disruption of *Mapk8^{LoxP}* and *Mapk9^{LoxP}* alleles in the vascular endothelium during embryonic development prior to analysis of FAL in adults and examination of blood flow by laser doppler imaging. (**F**) Quantitation of laser doppler analysis of limb blood flow demonstrated significantly enhanced blood perfusion blockade in adult mice with embryonic endothelial JNK-deficiency compared with control mice (mean ± SEM; n = 5~6 mice per group). (**G**) The adductor muscle vasculature of two E16.5 embryos obtained from a pregnant female mouse that was treated with tamoxifen at 12.5 dpc was examined by confocal microscopy. The *Rosa26^{mTmG}* genetic background allows detection of *Cre*-mdiated recombination in vascular endothelial cells (green). The data presented are representative of six mice examined. Source data are included as *Figure 4—source data 1*.

The following source data is available for figure 4:

*Figure 4 continued on next page*

*Figure 4 continued*

**Source data 1.** Source data for *Figure 4*.

The defect in blood flow of E$^{3KO}$ mice post-FAL could be mediated by cardiovascular dysfunction, but no changes in blood pressure, heart rate, or echocardiographic measurements of cardiac function were detected (*Figure 1—figure supplement 7A–C*). Moreover, contraction and endothelium-dependent relaxation responses in aortic explants from E$^{3KO}$ and control mice were similar (*Figure 1—figure supplement 7D*). These data indicate that neither cardiovascular dysfunction nor defective vasodilatory responses contribute to the post-FAL phenotype of E$^{3KO}$ mice.

## JNK is required for development of functional collateral arteries

The early and severe blood perfusion blockade in E$^{3KO}$ mice post-FAL suggests a defect in collateral artery function. Two highly stereotypic superficial arteries (gracilis collaterals) extend along the gracilis muscle in the medial aspect of the thigh (*Figure 1B*). Gracilis collaterals were identified as two lumenized continuous arteries that connected the PCFA to the saphenous artery (SA) (*Figure 1H*) and expanded radially during the post-FAL response (*Figure 1H,I*). In contrast, these arteries were abnormal in E$^{3KO}$ mice; arteries emerged from the PCFA and SA (*Figure 1H*), but were thin and branched into multiple smaller vessels forming a disorganized network (*Figure 1H*). Micro-computed tomography (μCT) analysis confirmed reduced collateral artery size and continuity in the limbs of E$^{3KO}$ mice (*Figure 1—figure supplement 8A*). These collateral artery defects may contribute to decreased blood flow and increased hypoxia in post-FAL E$^{3KO}$ mice, despite no overall reduction in muscle vascularization or macrophage recruitment (*Figure 1—figure supplement 8B–E*).

## Protection from ischemia following arterial occlusion requires MLK – JNK signaling

We tested whether disruption of genes that encode other JNK pathway components caused a similar post-FAL phenotype. The MLK group of MAP3K causes activation of the JNK pathway by a Rac1/Cdc42-dependent mechanism (*Gallo and Johnson, 2002*; *Kant et al., 2011*). Gene expression analysis demonstrated that *Map3k10* and *Map3k11* were the most highly expressed members of this group in endothelial cells (*Figure 2A*). Indeed, *Map3k10*$^{-/-}$ *Map3k11*$^{-/-}$ MLEC exhibited reduced phosphorylation of the JNK substrate cJun compared with control MLEC (*Figure 2B*). We therefore examined the post-FAL response of *Map3k10*$^{-/-}$ *Map3k11*$^{-/-}$ mice. Similar to E$^{3KO}$ mice, MLK-deficient mice showed increased blood flow blockade, failure of blood flow restoration by day 3 post-FAL, and necrosis (*Figure 2C–G*). Moreover, we found abnormal gracilis collateral arteries in *Map3k10*$^{-/-}$ *Map3k11*$^{-/-}$ mice (*Figures 2H* and *3A*). These data demonstrate that the MLK-JNK signaling pathway in endothelial cells is important for collateral artery patterning and the post-FAL response.

## MLK – JNK signaling is required for collateral artery development

The abnormal gracilis collateral arteries in MLK and JNK-deficient mice suggested that the JNK pathway could be important for collateral artery function, but these observations may also reflect a required role for the MLK-JNK pathway during development. To distinguish between these possiblities, we established mice with tamoxifen-inducible gene ablation. We found that compound *Mapk8/9/10* gene ablation in adult mice did not alter the post-FAL response (*Figure 4A–D*). In contrast, compound *Mapk8/9/10* gene ablation in embryos prior to FAL in adult mice caused increased blood flow blockade post FAL and failure to recover (*Figure 4E–G*). These data demonstrate that the post-FAL phenotype of E$^{3KO}$ mice is caused by an early developmental defect.

The pial collateral circulation that interconnects the distal branches of the middle cerebral and the anterior cerebral arteries has been studied (*Chalothorn and Faber, 2010*; *Lucitti et al., 2012*), but the formation of collateral arteries in muscle is unclear. We found that the gracilis collaterals in post-natal day 6 (P6) and P0 control mice interconnected the PCFA and the SA (*Figure 5A,B*). DiI perfusion analysis demonstrated that the arteries had lumens (*Figure 5A,B*) and were fully covered by smooth muscle cells at P6 (*Figure 5A*; SMA), but not at P0 (*Figure 5B*; SMA). In contrast, gracilis

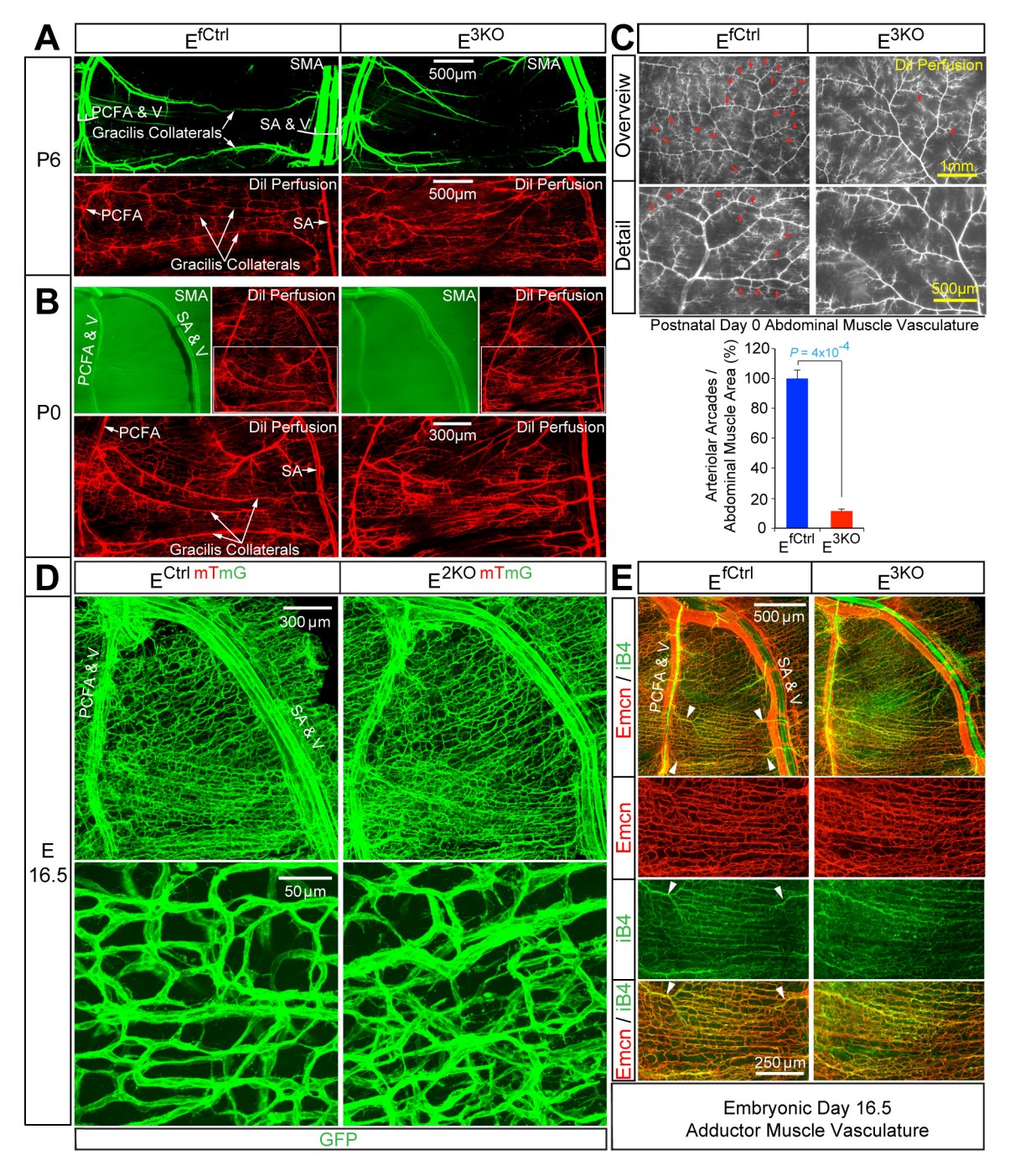

**Figure 5.** Endothelial JNK-deficient mice display abnormal native collateral arteries. (**A,B**) Representative confocal images (n = 7 mice) of whole mount adductor muscle vasculature reveals SMA-covered gracilis collateral arteries in P6 control mice, but not JNK-deficient mice (**A**). Confocal imaging of DiI perfused P6 adductor muscle vasculature (n = 5 mice) demonstrates distinct gracilis collaterals interconnecting the PCFA to the SA in control mice. Vessels emerging from the PCFA and the SA in E[3KO] mice do not fully interconnect, but branch into smaller vessels. At P0, gracilis collaterals were not SMA-covered, but were perfused with DiI in control mice (**B**). The analogous vessels in E[3KO] mice did not interconnect, but branched extensively into smaller vessels (**B**) (n = 5 mice). (**C**) Representative stereomicroscope images of DiI-perfused abdominal muscle arterial vasculature of control and JNK-deficient P0 mice. Arteriole-to-arteriole arcades (indicated by red arrows) were quantitated (mean ± SEM; n = 3 mice). (**D**) Representative confocal images (n = 3~4 mice) of whole mount adductor muscles of control and JNK-deficient E16.5 embryos showing GFP-labeled vascular endothelial cells. (**E**) Representative confocal images (n = 3~5 mice) of control and JNK-deficient E16.5 embryo adductor muscle vasculature immunostained for

*Figure 5 continued on next page*

*Figure 5 continued*

Endomucin (Emcn, red) and isolectinB4 (iB4, green). Prominent vessels emerging from the PCFA and SA are indicated with white arrowheads. Source data are included as *Figure 5—source data 1*.

The following source data and figure supplement are available for figure 5:

**Source data 1.** Source data for *Figure 5*.

**Figure supplement 1.** Intimate association of gracilis collaterals and peripheral nerves in adductor muscles.

collaterals in E$^{3KO}$ mice were not formed at P6 or P0 (*Figure 5A,B*). Individual vessels did emerge from the PCFA and the SA, but instead of interconnecting to form collaterals, these vessels branched off into multiple smaller caliber vessels (*Figure 5A,B*; Dil Perfusion) that lacked smooth muscle coverage at P6 (*Figure 5A*; SMA) and appeared to continue into the capillary circulation (*Figure 5A,B*; Dil Perfusion). Similarly, analysis of the abdominal muscle arterial circulation in P0 pups revealed numerous arteriolar arcades (direct arteriole-to-arteriole interconnections) in control mice, but this arterial patterning was significantly reduced in E$^{3KO}$ mice (*Figure 5C*). These defects in gracilis collateral and abdominal arteriolar arcade development were also detected in MLK-deficient mice (*Figure 3A,B*).

## Gracilis collateral development in control and JNK-deficient mice

To gain insight into the mechanism that might account for these defects in collateral artery patterning/maturation in E$^{3KO}$ mice, we examined the vasculature in whole mount preparations of adductor muscles in embryonic day 16.5 (E16.5) embryos. While large caliber vessels, including the FA/SA and PCFA, were established (*Figure 5D,E*), distinct collaterals directly interconnecting the PCFA and SA were not formed at E16.5 and the gracilis muscle was covered by a capillary plexus (*Figure 5D,E*). Prominent vessels do emerge from the PCFA and the SA (*Figure 5E*), but did not extend along the gracilis muscle as distinct collaterals; instead, these vessels branched and appeared to continue into the capillary plexus (*Figure 5E*). Gracilis collaterals may therefore form through a plexus intermediate. Remodeling of vessels within this plexus likely leads to the formation of collateral arteries in close apposition to nerve fibres (*Figure 5—figure supplement 1*). This process of maturation appears to start at the two distal ends, where the future gracilis collaterals emerge from the PCFA and the SA (*Figure 5E*, arrowheads) and continue toward the middle of the muscle; a pattern of remodeling that likely reflects the blood flow characteristics of these vessels (*Meisner et al., 2013*). Studies of E16.5 E$^{3KO}$ embryos demonstrated that the gracilis muscle capillary plexus was hyperbranched, denser, and disorganized with vessels of variable width that elaborated more filopodia than control embryos (*Figure 5D,E*). This observation suggests that the failure of collateral artery formation in E$^{3KO}$ mice may be caused by defective sprouting angiogenesis that initially generates a hyperbranched, denser, and more chaotically organized plexus that fails to properly remodel.

## JNK regulates vessel sprouting during developmental angiogenesis

To test whether JNK-deficiency caused endothelial cell hypersprouting, we examined retinal vascular development during the early postnatal period because this is a well-characterized system that enables analysis of sprouting angiogenesis in a vascular plexus that initially extends from the center towards the periphery of the retina in two dimensions (*Eilken and Adams, 2010*). Analysis of retinal flatmounts from P6 E$^{3KO}$ mice demonstrated significantly reduced radial extension of the vascular plexus (*Figure 6A–C,L*). Closer examination demonstrated higher vascular density in the growing angiogenic front of the mutant retinas compared to littermate control mice (*Figure 6D–G,H,J,M*). Vascular extension in the retina occurs through the coordinated interaction, migration, and proliferation of endothelial tip and stalk cells together with non-endothelial cells, including pericytes, that stabilize the vascular plexus. We found no differences in vessel pericyte coverage (*Figure 6—figure supplement 1*); however, the angiogenic front of the mutant retinas included larger numbers of tip cells and a larger number of filopodia (*Figure 6H–K,N,O*). Similar data were obtained from analysis of E$^{2KO}$ and *Map3k10$^{-/-}$ Map3k11$^{-/-}$* mice (*Figure 6—figure supplements 2* and *3*). These data

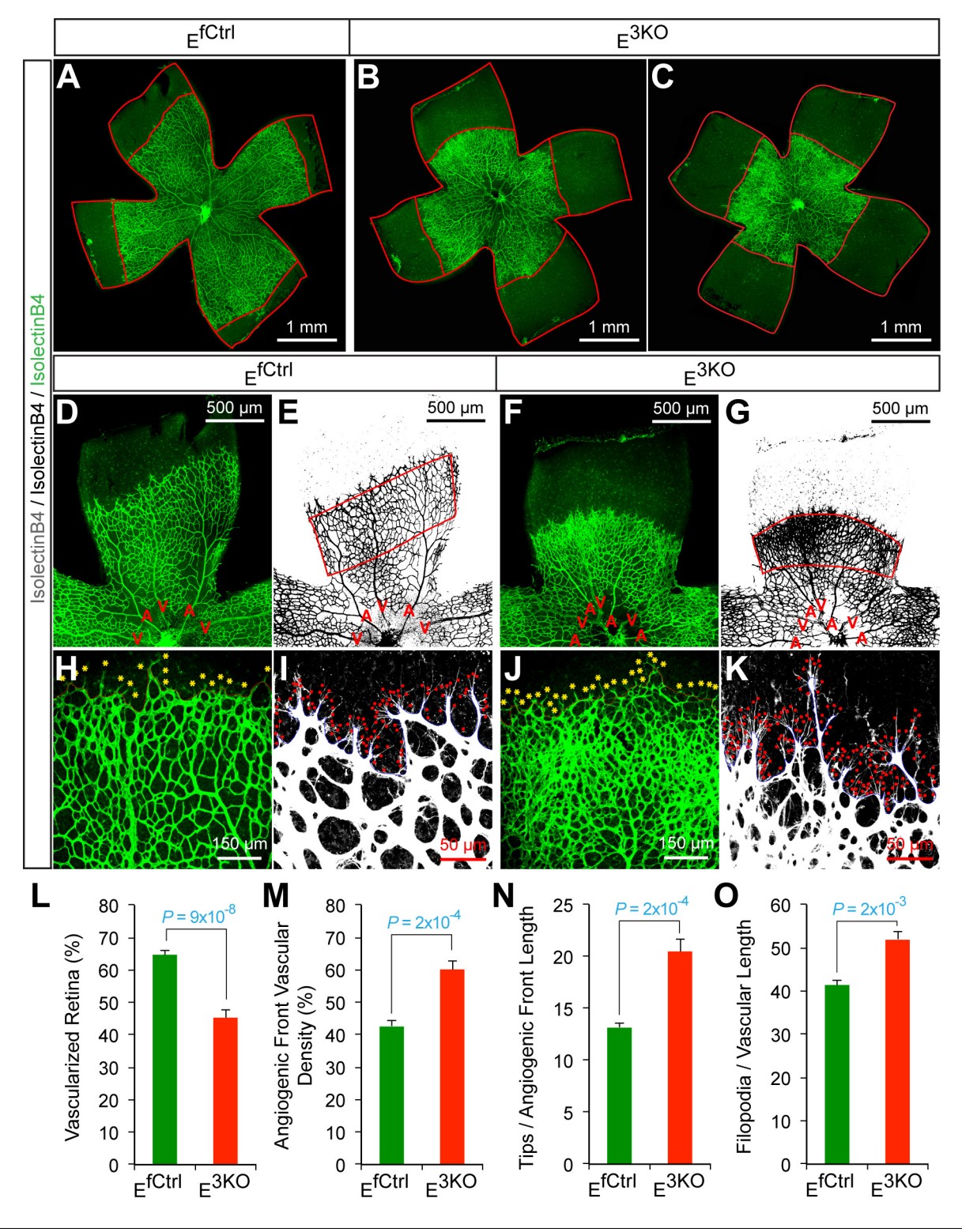

**Figure 6.** Abnormal retinal vascular development associated with excessive sprouting of tip cells in endothelial JNK-deficient mice. (A–C) Collages of confocal images of P6 whole mount retinas stained with isolectinB4 (iB4) show reduced vascular extension in JNK-deficient retinas (B,C) compared with littermate control retinas (A). The images are representative of 17~31 retinas examined for each genotype. (D–K) Higher magnification reveals increased vascular density (D–G, H, J), increased numbers of tip cells (yellow asterisks, H & J) and increased numbers of filopodia (red dots, I and K) at the vascular front region of JNK-deficient retinas compared to littermate control retinas. A, artery; V, vein. (L–O) The vascularized retinal area (L), vascular

*Figure 6 continued on next page*

*Figure 6 continued*

density within angiogenic front regions outlined in panels **E** & **G** (**M**), tip cell number (**N**), and filopodia number (**O**) is presented (mean ± SEM; n = 17~31 (panel I); n = 6~10 (panels **M–O**). Source data are included as *Figure 6—source data 1*.

The following source data and figure supplements are available for figure 6:

**Source data 1.** Source data for *Figure 6*.

**Figure supplement 1.** NG2$^+$ pericyte coverage of the P6 retinal vasculature.

**Figure supplement 2.** Abnormal retinal vascular development associated with excessive sprouting in endothelial JNK1/2-deficient mice.

**Figure supplement 2—source data 1.** Source data for *Figure 6—figure supplement 2*.

**Figure supplement 3.** Abnormal retinal vascular development associated with excessive sprouting in *Map3k10$^{-/-}$ Map3k11$^{-/-}$* mice.

**Figure supplement 3—source data 1.** Source data for *Figure 6—figure supplement 3*.

confirm that the MLK-JNK signaling pathway restrains excessive endothelial cell sprouting during developmental angiogenesis.

## JNK promotes signaling by the Dll4 – Notch pathway

To examine the molecular mechanisms that contribute to hypersprouting caused by defects in the MLK-JNK pathway, we examined gene expression by primary endothelial cells isolated from control and E$^{3KO}$ mice. We found 781 genes that were differentially expressed between E$^{3KO}$ and control endothelial cells. Gene ontology analysis demonstrated significant enrichment for several biological processes, including vascular development and morphogenesis, and we identified 64 differentially expressed genes that might contribute to vascular defects (*Figure 7—figure supplement 1*). These genes included components of the Notch signaling pathway; for example, *Dll4, Hey1, Hes1*, and *Lfng* (*Figure 7A,B*). This may be significant because the Notch pathway plays a major role during developmental angiogenesis, including tip/stalk cell specification and endothelial cell sprouting (*Roca and Adams, 2007*; *Phng and Gerhardt, 2009*). Indeed, the hypersprouting defects observed in E$^{2KO}$, E$^{3KO}$ and *Map3k10$^{-/-}$ Map3k11$^{-/-}$* mice resemble those previously reported for mice with reduced Dll4/Notch signaling (*Hellström et al., 2007*; *Suchting et al., 2007*; *Benedito et al., 2009*), including *Lfng$^{-/-}$* mice (*Benedito et al., 2009*). Moreover, *Dll4$^{-/+}$* mice display perturbations in collateral artery formation and, like *Notch1$^{-/+}$* mice (*Takeshita et al., 2007*), show reduced recovery of blood perfusion in models of vascular occlusion (*Cristofaro et al., 2013*). To test whether JNK-deficiency regulates Notch signaling in endothelial cells, we examined the effect of angiogenic cytokines that can induce expression of the Notch ligand Dll4 and engage the Notch signaling pathway. Treatment of endothelial cells with VEGF or bFGF caused increased expression of Dll4 protein and promoted accumulation of the Notch intracellular domain (NICD). However, these responses were suppressed in JNK-deficient endothelial cells (*Figure 7C,D*). Indeed, reduced Dll4 protein expression by E$^{3KO}$ endothelial cells was observed in vitro and in vivo (*Figure 7B–E*). These data indicate that JNK can promote Notch signaling in endothelial cells by regulating Dll4 expression.

## Discussion

We have established mice with JNK signaling defects in the vascular endothelium to study the role of JNK in vascular function. We found that JNK is not required for angiogenesis or arteriogenesis in adult mice. However, JNK plays an essential role during developmental vascular morphogenesis, including the formation of native collateral arteries that provide an alternate route for blood flow and serve to protect against ischemic tissue damage. This action of JNK is mediated by the MLK signaling pathway. Defects in the MLK-JNK pathway result in the loss of muscle collateral circulation and profoundly suppress protective responses to arterial occlusion.

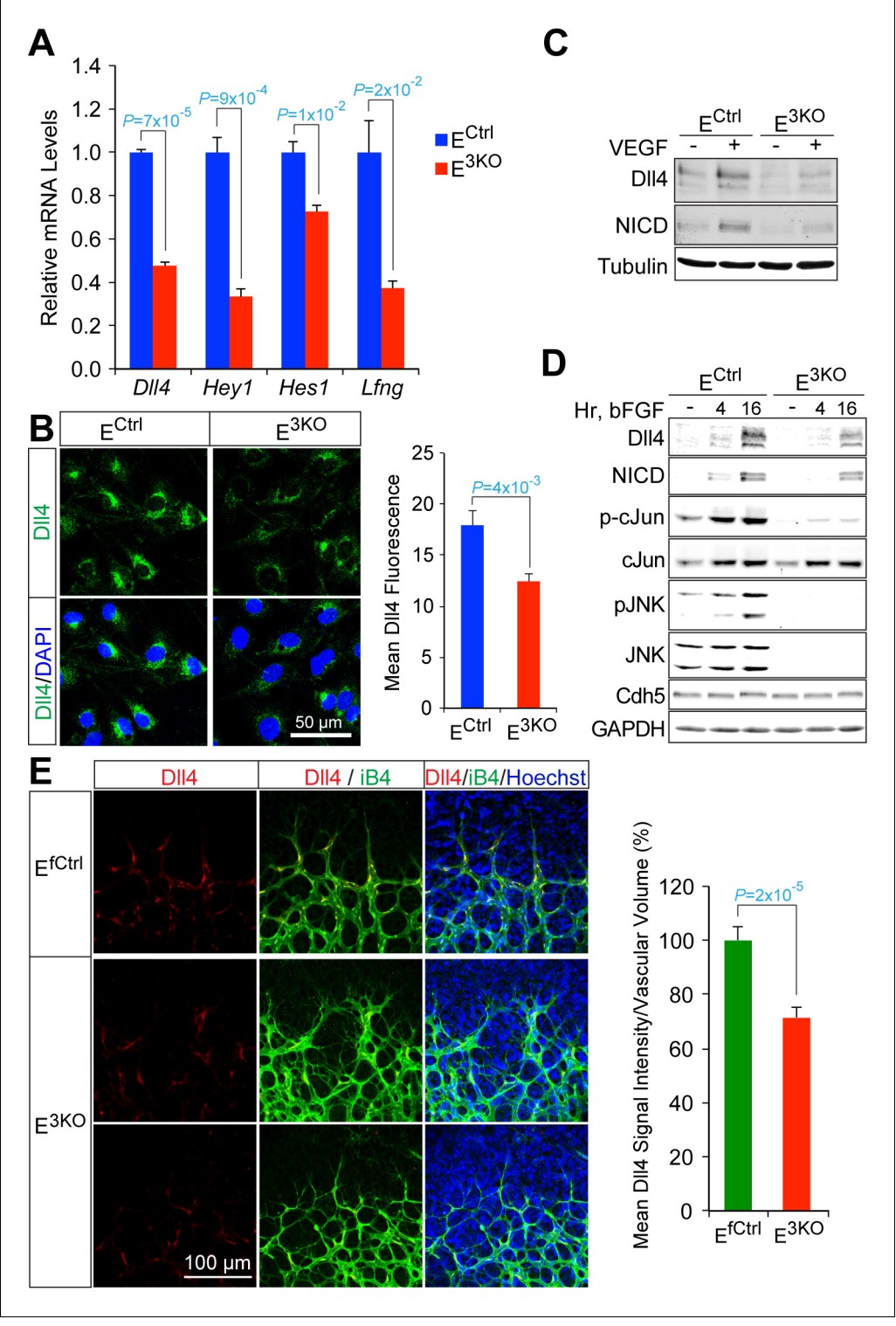

**Figure 7.** Reduced Dll4 / Notch signaling in the JNK-deficient vascular endothelium. (**A**) Quantitative RT-PCR analysis of Notch pathway genes revealing reduced expression in JNK-deficient primary endothelial cells compared with control cells (mean ± SEM; n = 4). The data shown are representative of the results obtained with three independent primary endothelial cell preparations. (**B**) Dll4 expression by control and JNK-deficient primary endothelial cells was examined by Immunofluorescence analysis (mean ± SEM; n = 10). (**C**) Control and JNK-deficient primary endothelial cells treated without and with 100 ng/ml VEGF (16 hr) were examined by immunoblot analysis by probing with antibodies to Dll4, Notch intracellular domain (NICD), and αTubulin. The data are

*Figure 7 continued on next page*

*Figure 7 continued*
representative of experiments performed using two independent endothelial cell preparations. (**D**) Endothelial cells treated without and with 100 ng/ml bFGF were examined by immunoblot analysis by probing with antibodies to Dll4, NICD, pSer[63]-cJun, cJun, pJNK, JNK, Cdh5 and GAPDH. The data are representative of experiments performed using two independent endothelial cell preparations. (**E**) Confocal immunofluorescence analysis of P6 whole mount retinas immunostained for Dll4 (red), isolectinB4 (iB4, green), and Hoechst (DNA, blue) demonstrates that JNK-deficiency causes reduced expression of Dll4 at the angiogenic vascular front compared with retinas from littermate control mice (mean ± SEM; n = 42~44). Source data are included as *Figure 7—source data 1*.
The following source data and figure supplements are available for figure 7:
**Source data 1.** Source data for *Figure 7*.
**Figure supplement 1.** RNA-Seq analysis of differentially expressed genes between control and JNK-deficient endothelial cells.
**Figure supplement 1—source data 1.** Source data for *Figure 7—figure supplement 1*.

## Collateral artery development

Studies of the development of leptomeningeal (or pial) collaterals that interconnect the medial, anterior and posterior cerebral artery trees in the brain have provided evidence that these collaterals form during embryonic development starting at ~E13.5 as sprout-like extensions of endothelial cells from arterioles of existing cerebral artery trees (*Lucitti et al., 2012*). These nascent vessels appear to course above the pial capillary plexus and fuse with an arteriole from an adjacent arterial tree (*Lucitti et al., 2012*). By E15.5, a portion of these collaterals have acquired expression of the arterial marker EphrinB2 (*Chalothorn and Faber, 2010*). Pial collateral density peaks ~E18.5 and is followed by extensive remodeling, maturation and pruning that continues postnatally, achieving adult form and density by P21 (*Chalothorn and Faber, 2010*). The process of collateral artery formation during embryonic development has been termed collaterogenesis (*Chalothorn et al., 2009*; *Faber et al., 2014*).

Little is known about the development of the collateral arteries in muscle. Our studies focused on the development of the gracilis collateral arteries. We found that a vascular plexus is formed between the proximal caudal femoral artery (PCFA) and the saphenous artery (SA) at ≤ E16.5 before gracilis collaterals are detected at ≤ P0 and covered with smooth muscle at ≤ P6 (*Figure 5*). This observation suggests that gracilis collaterals may be formed through a plexus intermediate by selection and maturation within a pre-formed capillary network that separates adjacent arteries. The presence of nerve fibres may contribute to this process (*Figure 5—figure supplement 1*).

Developmental defects in gracilis collateral development were observed in mice with JNK deficiency in endothelial cells. The vascular plexus between adjacent arteries at E16.5 in JNK-deficient mice is denser and more chaotically organized (*Figure 5*). Highly variable vessel thickness and increased numbers of filopodia are also evident in the JNK-deficient vasculature (*Figure 5*). Similar defects were observed in the developing retinal vasculature (*Figure 6*). At P0, continuous and distinct gracillis collateral vessels interconnecting the PCFA to the SA were fully formed in control mice, but the analogous vessels in the gracillis muscle of JNK-deficient mice were defective with extensive branching into smaller vessels that appeared to enter the capillary circulation. These observations indicated that an endothelial cell patterning defect may account for the absence of gracillis collateral arteries in the JNK-deficient mice. Hypersprouting represents one aspect of this patterning defect (*Figure 6*) and reduced Dll4 – Notch signaling may significantly contribute to this phenotype (*Figure 7*).

## VEGF and Notch signaling in JNK-deficient vascular endothelium

The formation of properly organized vascular networks is essential for function and requires the coordinated interaction of numerous factors and signaling pathways that regulate diverse cellular processes. VEGF signaling promotes endothelial cell survival, proliferation and motility while Dll4–Notch signaling suppresses VEGF signaling, in part, by regulating VEGFR expression. These

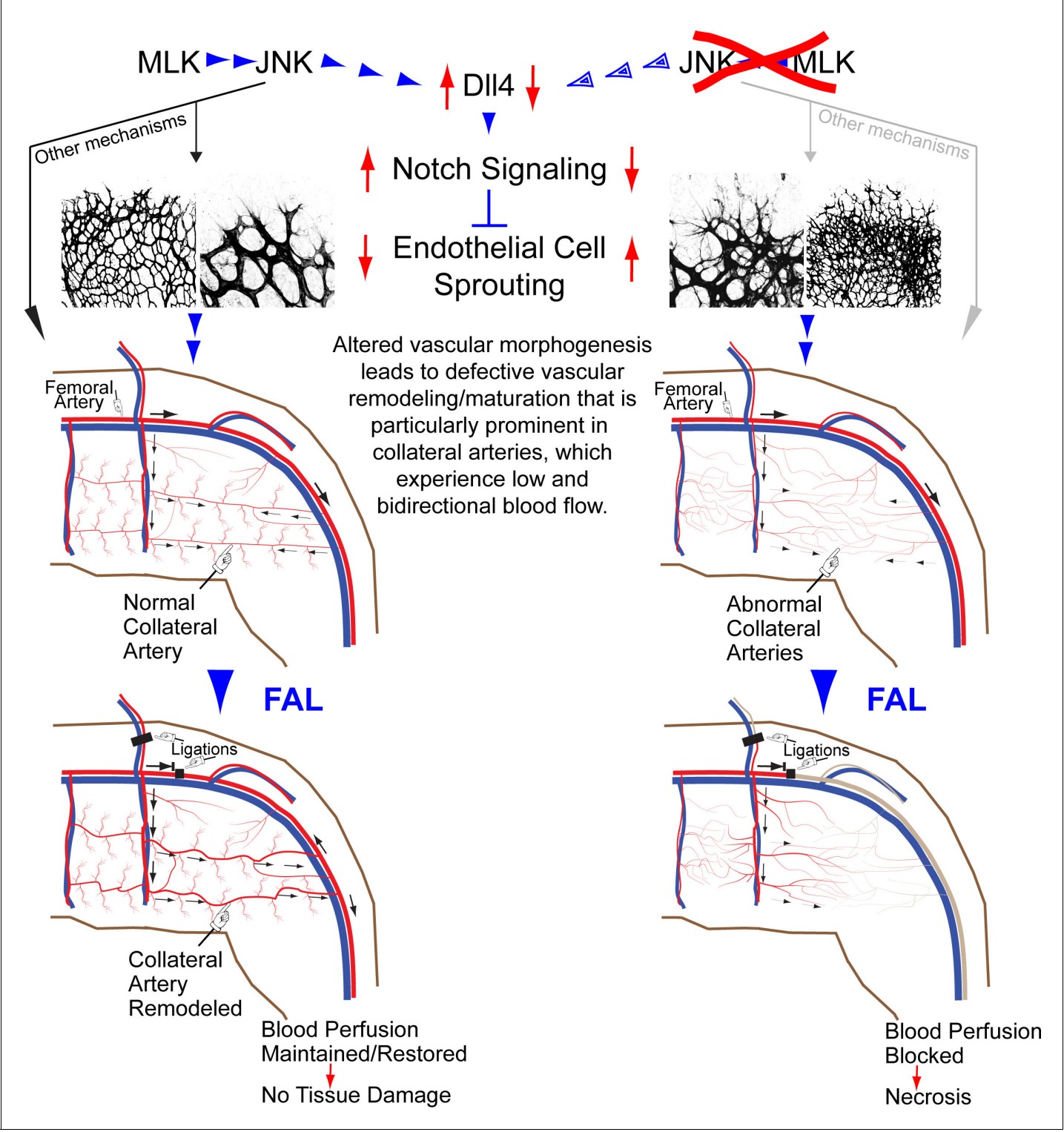

**Figure 8.** Schematic illustration of the role of the MLK-JNK pathway in vascular morphogenesis, formation of native collateral arteries, and the femoral artery ligation (FAL) model of hindlimb ischemia.

interactions between VEGF and Notch signaling underly the specification of endothelial cell phenotypes during angiogenesis, including highly motile tip cells that extend numerous filopodia and trailing stalk cells with low motility that form the lumen of nascent tubules (*Roca and Adams, 2007*; *Phng and Gerhardt, 2009*; *Eilken and Adams, 2010*; *Jakobsson et al., 2010*). The proper specification and interplay of tip and stalk cells is essential for the orchestration of sprouting angiogenesis that mediates expansion of vascular networks. Moreover, VEGF and Notch signaling are also critical for formation of the native collateral circulation (*Cristofaro et al., 2013*).

Studies of retinal vascular development demonstrate that MLK – JNK signaling regulates tip cell identity, and filopodia dynamics (*Figure 6*). Consequently, defects in the MLK – JNK signaling pathway cause excessive sprouting angiogenesis characterized by an increased number of tips and filopodia, increased vascular density, and decreased expression of the Notch ligand Dll4 at the angiogenic front (*Figure 7*). Excessive sprouting angiogenesis may contribute to the critical requirement for MLK – JNK signaling during native collateral artery development (*Figure 5*) and this phenotype may result from promotion of Dll4 expression by MLK – JNK signaling (*Figure 7*). Indeed, it is established that defects in Notch signaling, including loss of *Dll4* (*Hellström et al., 2007*), *Notch1* (*Hellström et al., 2007*), *Lfng* (*Benedito et al., 2009*), *RBP-J* (*Dou et al., 2008*; *Izumi et al., 2012*) or chemical inhibition of γ-secretase (*Hellström et al., 2007*), cause excessive endothelial cell sprouting during retinal vasculature development. Moreover, *Dll4$^{+/-}$* mice display defects in blood perfusion restoration following femoral artery ligation (*Cristofaro et al., 2013*). These data support the conclusion that the decreased Dll4 expression caused by JNK-deficiency in the vascular endothelium promotes excessive sprouting angiogenesis that disrupts the development of muscle collateral vessels (*Figure 8*).

## Conclusions

Our study points to an important role for JNK-mediated regulation of Dll4 – Notch signaling and vascular morphogenesis, and identifies a MLK – JNK signaling axis that is critical for native collateral artery formation. Disruption of this pathway causes defective collateral artery formation that results in severe blood perfusion blockade and tissue injury following arterial occlusion. Our study provides insight into the mechanism that controls muscle collaterogenesis, which is critically important for the response to arterial occlusive disease. Moreover, our analysis suggests that human germ-line mutations in JNK pathway genes may contribute to an increased risk for poor outcomes in patients following exposure to an ischemic insult.

# Materials and methods

## Mice

C57BL/6J mice (RRID:IMSR_JAX:000664), B6.SJL-*Ptprc$^a$ Pepc$^b$/BoyJ* mice (RRID:IMSR_JAX:002014), B6.129P2-*Lyz2$^{tm1(cre)Ifo}$*/J (also known as *Lyz2-Cre mice and* Φ$^{Ctrl}$ *mice*; RRID:IMSR_JAX:004781) (*Clausen et al., 1999*), B6.FVB-Tg(Cdh5-cre)7Mlia/J mice (RRID:IMSR_JAX:006137) (*Alva et al., 2006*), B6.Cg-Tg(Vav1-cre)A2Kio/J mice (RRID:IMSR_JAX:008610) (*de Boer et al., 2003*), B6.FVB (129S4)Tg(Ckmm-cre)5Khn/J mice (RRID:IMSR_JAX:006475) (*Brüning et al., 1998*), and B6.129(Cg)-*Gt(ROSA)26Sor$^{tm4(ACTB-tdTomato,-EGFP)Luo}$*/J mice (also known as *Rosa26$^{mTmG}$* mice) (RRID:IMSR_JAX: 007676) (*Muzumdar et al., 2007*) were obtained from The Jackson Laboratories (Bar Harbor, ME). Tg(Cdh5-cre/ERT2)1Rha mice (RRID:IMSR_TAC:13073) (*Wang et al., 2010*) were provided by Prof. Ralf H. Adams. We have previously described *Mapk8$^{LoxP/LoxP}$*, *Mapk9$^{LoxP/LoxP}$*, *Mapk8$^{-/-}$* (RRID: IMSR_JAX:004319), *Mapk9$^{-/-}$* (RRID:IMSR_JAX:004321), *Mapk10$^{-/-}$* (RRID:IMSR_JAX:004322), Φ$^{KO}$ mice (*Lyz2-Cre$^+$ Mapk8$^{LoxP/LoxP}$ Mapk9$^{LoxP/LoxP}$*), and *Map3k10$^{-/-}$ Map3k11$^{-/-}$* mice (RRID:MGI: 5296041) (*Dong et al., 1998*; *Yang et al., 1998*; *Kuan et al., 2003*; *Das et al., 2007*; *Kant et al., 2011*; *Han et al., 2013*). We generated the following mice:

E$^{3KO}$ (*Cdh5-Cre$^{+/-}$ Mapk8$^{LoxP/LoxP}$ Mapk9$^{LoxP/LoxP}$ Mapk10$^{-/-}$*)
E$^{fCtrl}$ (*Cdh5-Cre$^{-/-}$ Mapk8$^{LoxP/LoxP}$ Mapk9$^{LoxP/LoxP}$ Mapk10$^{-/-}$*)
E$^{Ctrl}$ (*Cdh5-Cre$^{+/-}$ Mapk8$^{+/+}$ Mapk9$^{+/+}$ Mapk10$^{-/-}$*)
E$^{2KO}$ (*Cdh5-Cre$^{+/-}$ Mapk8$^{LoxP/LoxP}$ Mapk9$^{LoxP/LoxP}$*)
E$^{LoxP}$ (*Cdh5-Cre$^{-/-}$ Mapk8$^{LoxP/LoxP}$ Mapk9$^{LoxP/LoxP}$*)
E$^{WT}$ (*Cdh5-Cre$^{+/-}$ Mapk8$^{+/+}$ Mapk9$^{+/+}$*)

E$^{2KO:mTmG}$ (Cdh5-Cre$^{+/-}$ Mapk8$^{LoxP/LoxP}$ Mapk9$^{LoxP/LoxP}$ Rosa26$^{mTmG+/-}$)
E$^{mTmG}$ (Cdh5-Cre$^{+/-}$ Rosa26$^{mTmG+}$)
iE$^{3KO}$ (Cdh5-Cre/ERT2$^{+/-}$ Mapk8$^{LoxP/LoxP}$ Mapk9$^{LoxP/LoxP}$ Mapk10$^{-/-}$)
iE$^{fCtrl}$ (Cdh5-Cre/ERT2$^{-/-}$ Mapk8$^{LoxP/LoxP}$ Mapk9$^{LoxP/LoxP}$ Mapk10$^{-/-}$)
iE$^{Ctrl}$ (Cdh5-Cre/ERT2$^{+/-}$ Mapk8$^{+/+}$ Mapk9$^{+/+}$ Mapk10$^{-/-}$)
iE$^{2KO:mTmG}$ (Cdh5-Cre/ERT2$^{+/-}$ Mapk8$^{LoxP/LoxP}$ Mapk9$^{LoxP/LoxP}$ Rosa26$^{mTmG+/-}$)
E$^{LoxP:mTmG}$ (Cdh5-/CreERT2$^{-/-}$ Mapk8$^{LoxP/LoxP}$ Mapk9$^{LoxP/LoxP}$ Rosa26$^{mTmG+/-}$)
iE$^{mTmG}$ (Cdh5-Cre/ERT2$^{+/-}$ Mapk8$^{+/+}$ Mapk9$^{+/+}$ Rosa26$^{mTmG+/-}$)
H$^{2KO}$ (Vav1-Cre$^{+/-}$ Mapk8$^{LoxP/LoxP}$ Mapk9$^{LoxP/LoxP}$)
H$^{LoxP}$ (Vav1-Cre$^{-/-}$ Mapk8$^{LoxP/LoxP}$ Mapk9$^{LoxP/LoxP}$)
H$^{WT}$ (Vav1-Cre$^{+/-}$ Mapk8$^{+/+}$ Mapk9$^{+/+}$)
M$^{2KO}$ (Ckm-Cre$^{+/-}$ Mapk8$^{LoxP/LoxP}$ Mapk9$^{LoxP/LoxP}$)
M$^{WT}$ (Ckm-Cre$^{+/-}$ Mapk8$^{+/+}$ Mapk9$^{+/+}$)

All mice used in this study were backcrossed ($\geq$ ten generations) to the C57BL/6J strain. The mice were housed in a specific pathogen-free facility accredited by the American Association for Laboratory Animal Care. The animal studies were approved by the Institutional Animal Care and Use Committees of the University of Massachusetts Medical School, Tufts University School of Medicine, and Brigham and Women's Hospital.

## Genotyping

PCR assays with genomic DNA and the amplimers 5'-TTACTGACCGTACACCAAATTTGCCTGC-3' and 5'-CCTGGCAGCGATCGCTATTTTCCATGAGTG-3' were used to detect the Cre$^+$ allele (450 bp). The amplimers 5'CCTCAGGAAGAAAGGGCTTATTTC-3' and 5'-GAACCACTGTTCCAATTTCCATCC-3' detected the Mapk8$^+$ allele (1550 bp), the Mapk8$^{LoxP}$ allele (1,095 bp), and the Mapk8$^\Delta$ allele (395 bp). The amplimers 5'-GTTTTGTAAAGGGAGCCGAC-3' and 5'-CCTGACTACTGAGCCTGGTTTCTC-3' were used to detect the Mapk9$^+$ allele (224 bp) and the Mapk9$^{LoxP}$ allele (264 bp). The amplimers 5'-GGAATGTTTGGTCCTTTAG-3', 5'-GCTATTCAGAGTTAAGTG-3', and 5'-TTCATTCTAAGCTCAGACTC-3' were used to detect the Mapk9$^{LoxP}$ allele (560 bp) and the Mapk9$^\Delta$ allele (400 bp). The amplimers 5'-CCTGCTTCTCAGAAACACCCTTC-3', 5'-CGTAATCTTGTCACAGAAATCCCATAC-3' and 5'-CTCCAGACTGCCTTGGGAAAA-3' were used to detect the Mapk10$^+$ allele (437 bp) and the Mapk10$^-$ allele (250 bp). The amplimers 5'-CTCTGCTGCCTCCTGGCTTCT-3', 5'-CGAGGCGGATCACAAGCAATA-3' and 5'-TCAATGGGCGGGGGTCGTT-3' were used to detect the mTmG allele (250 bp) and the WT allele (330 bp). The amplimers 5'-CCTGGTTCTCACTGGGACAACAG-3', 5'-GTCACATCCACTTTCCTGGGC-3', and 5'-CGCCTTCTATCGCCTTCTTGA-3' detected the Map3k10$^+$ allele (500 bp) and the Map3k10$^-$ allele (600 bp). The amplimers 5'-AGCAAACTCCGAGCAAGGGAC-3', 5'-GGCTAAACCAGAACTCAAGCGTG-3', and 5'-GTAGAAGGTGGCGCGAAGGG-3' were used to detect the Map3k11$^+$ allele (160 bp) and the Map3k11$^-$ allele (280 bp).

## Tamoxifen

Male mice (6–8 wk old) were treated with tamoxifen (Sigma-Aldrich (St. Louis, MO)) dissolved in 2% ethanol / 98% sunflower seed oil by intraperitoneal injection (1 mg/mouse) 5 times (at 48 hr intervals). Mouse embryos were treated by oral gavage of Cre$^-$ female mice at 12.5 days post coitus (dpc) with 3 mg tamoxifen dissolved in 2% ethanol / 98% sunflower seed oil. The tamoxifen-treated pups were delivered by C-section at ~ 19.5 dpc and transferred to foster mothers.

## Femoral artery ligation (FAL) and laser doppler imaging

Unilateral FAL and laser doppler imaging was performed using 10–14 week old male mice as previously described (*Limbourg et al., 2009*; *Craige et al., 2011*) with the following modifications. Two ligation protocols were performed. In one protocol we ligated the femoral artery at its origin. The second protocol involved ligation of the femoral artery between the proximal caudal femoral artery and the popliteal artery as well as ligation of the superficial epigastric artery. The second ligation schema allows for more blood flow to be diverted to the gracillis collateral circulation. Quantitative scores for ischemia and movement post-FAL were performed as described (*Chalothorn et al., 2007*).

## Aortic ring angiogenesis assays

The aortic ring assays were performed in collagen as previously described (*Baker et al., 2012*).

## Laser-induced choroidal neovascularization

Choroidal neovascularization was induced in mice using a 532 nm laser as previously described (*Cashman et al., 2011*). Four laser spots/eye were applied. The eyes were harvested 7 days post-injury, fixed in 4% paraformaldehyde at 4°C overnight and eyecups dissected and subjected to wholemount immunofluorescence analysis using a TCS SP2 Leica confocal microscope.

## Tumor angiogenesis assays

One million congenic B16F10 melanoma cells (American Type Culture Collection Cat# CRL-6475, RRID:CVCL_0159) were injected subcutaneously on both flanks of mice. Tumors were harvested 2 weeks later, weighed, imaged using a Zeiss Stereo Discovery V12 stereomicroscope, fixed in 4% paraformaldehyde (4°C, 12 hr), dehydrated sequentially in 15% and 30% sucrose in PBS, and embedded in Optical Cutting Temperature (OCT) prior to preparing frozen sections (10 µm). Sections were allowed to dry at room temperature, rehydrated in PBS, blocked and permeabilized in 10% normal donkey serum, 0.1% Triton X-100 in PBS for 1 hr at RT and incubated with primary antibodies; mouse anti-smooth muscle actin (1:500, Sigma-Aldrich Cat# F3777, RRID:AB_476977) and rat anti-CD31 (1:50, BD Biosciences (San Jose, CA) Cat# 558736, RRID:AB_397095) in 1% BSA PBS for 2 hr at RT. Sections were washed $3 \times 5$ min each with PBS and incubated with Alexa Fluor 546-goat anti mouse and Alexa Fluor 488-goat anti rat antibodies in 1% BSA PBS for 1 hr at RT. Following washing as above, DNA was stained with DAPI, sections mounted in FluoromountG (Southern Biotech (Birmingham, AL)) and imaged on a TCS SP2 Leica confocal microscope.

## Coronary artery ligation

Myocardial infarction studies were done at the Partners Cardiovascular Physiology Core at Brigham and Women's Hospital, as previously described (*Bauer et al., 2011*; *Li et al., 2011*). Briefly, adult male mice were anesthetized by IP injection of a mixture of ketamine (40 mg/kg) and xylazine (10 mg/kg), intubated, and mechanically ventilated. Following thoracotomy, the pericardium was removed, and the proximal left coronary artery was permanently occluded with an intramural stitch.

## Echocardiography

Echocardiography (Vevo 2100, VisualSonics Inc) was performed at the Partners Cardiovascular Physiology Core at Brigham and Women's Hospital as previously described (*Bauer et al., 2011*). Two-dimensional and M-mode echocardiographic images were obtained from lightly sedated (1% isoflurane in oxygen) mice and recorded. M-mode images were obtained from the parasternal short-axis view at the level of the papillary muscles and used for measurements.

## Blood pressure and heart rate

Blood pressure and heart rate measurements were done on 10–14 week old male mice using a non-invasive computerized tail cuff system (BP-2000, VisiTech Systems). Mice were trained for 1 week, and then systolic and diastolic blood pressure and heart rate were recorded as the mean of at least 16 successful measurements over 1 week.

## Measurement of arterial contraction / relaxation responses

Aortas were harvested from mice, flushed and cleaned of periaortic fat as described (*Baker et al., 2012*), cut into 2 mm long rings and equilibrated in Opti-MEM containing penicillin/streptomycin overnight at 37°C. Contraction and relaxation responses were measured using a 6-mL vessel myograph (Danish Myo Technology) as previously described (*36*) with the following modifications. Arterial contraction in response to increasing doses of phenylephrine (Phe) was recorded and expressed as percent of maximum contraction obtained in response to incubation in K-PSS (60 mM potassium-containing physiologic salt solution [mM: NaCl 130, KCl 4.7, KHPO$_4$ 1.18, MgSO$_4$ 1.17, CaCl$_2$ 1.6, NaHCO$_3$ 14.9, dextrose 5.5, CaNa$_2$/EDTA 0.03]). Vasorelaxation in response to increasing doses of acetylcholine was recorded following pre-contraction with Phe ($10^{-6}$ M).

## µCT analysis

Hindlimbs were scanned in air aligned axially on a Scanco (Wayne, PA) µCT 40 at 70kVp, 114 µA and a resolution of 10 µm. The region of interest (ROI) included the entire hindlimb. To obtain the bone/vasculature overlay image, a contour around the entire ROI was utilized and segmented to include all soft and hard tissue. A second contour of the same ROI with the bone removed was also performed and segmented. The segmentation parameters included the values 0.8 Gauss sigma, 1.0 Gauss support, and a threshold of 212–1000 (density range of 500 mg of HA/cm$^3$). The two segmented files were overlaid using Scanco's IPL Transparency program and a false color image of the resulting file was created using the 3D Display program.

## Microfil and bismuth/gelatin perfusions

Mice were anesthetized (150 mg/kg ketamine and 13 mg/kg xylazine) and treated by intravenous injection of 400 U heparin prior to thoracotomy. The right atrium was severed and the mice were maximally vasodilated by infusing, via the left ventricle, 30 ml normal saline containing 1 g/l adenosine, 4 mg/l papaverine, and 100 µg/ml heparin followed by 15 ml 2% formalin and ~0.5 ml uncatalyzed blue Microfil (Flow Tech) to help visualize the abdominal aorta during cannulation. Mice were then transected just below the diaphragm, the abdominal aorta was cannulated (Mc-28, Braintree Scientific (Braintreet, MA)) and the vasculature perfused using a syringe gun (IGSET-3510, Medco) with ~3 ml undiluted catalyzed blue Microfil or 10 ml of a warm 50% Bismuth (prepared as described [*Simons, 2008*]) / 7% gelatin in normal saline. The aorta and vena cava were then clamped and the perfusate was allowed to polymerize for at least an hour (4°C) before the hindlimbs were harvested, the skin removed, and the limbs placed in 10% formalin. We dissected the medial surface of adductor muscles of fixed, dehydrated (70 and 100% ethanol), and Microfil-perfused hindlimbs. The tissue was cleared in methyl salicylate (Sigma) and imaged with a stereomicrosope (Ziess).

## Dil perfusions

The Dil solution was prepared as previously described (*Li et al., 2008*), but with the addition of a filtration (40 µm) step to remove undissolved particles. P0 or P6 pups were euthanized by isofluorane inhalation, decapitated, and immediately perfused via the left ventricle with 3 or 5 ml, respectively, of Dil solution using a 10 ml syringe and 27 gauge needle and/or the thoracic aorta using a micro cannula (Mc-28, Braintree Scientific). Pups were then rinsed with PBS and fixed/stored in 4% paraformaldehyde (PFA) at 4°C until dissected.

## Ocular dissections

Eyes were fixed in 4% paraformaldehyde (RT,1 hr or 4°C, 12 hr) and retinas were dissected as previously described (*Pitulescu et al., 2010*).

## Muscle dissections

P6 pups were perfused with PBS via the left ventricle (and E16.5 and P0 pups were rinsed in PBS) prior to fixation in 4% paraformaldehyde (4°C, 12 hr). Using a steromicroscope, the mice were transected below the diaphragm and a mid-sagital incision was performed to separate the two hindlimbs and the associated abdominal musculature. The skin and associated adipose tissue was carefully removed and the abdominal muscles isolated via incisions at their attachment to the pelvis and vertebral column. The entire medial surface of the hindlimb adductor muscles was harvested en block via dissection 1–2 mm around the saphenous, femoral and proximal caudal femoral arteries. Muscle tissues were then either cleared sequentially (70% and 90% glycerol/PBS, at least 5 hr each) and mounted in 90% glycerol/PBS for direct visualization of GFP / Dil or processed for immunofluorescence analysis. A similar procedure was used to examine adult muscle.

## Whole mount lectin and immunofluorescence staining

Muscles were blocked and permeabilized in 1% BSA, 0.5% Triton X-100 PBS (12 hr, 4°C). Tissues were equilibrated by washing 3 × 10 min each with Pblec buffer (1% Triton X-100, 1 mM CaCl$_2$, 1 mM MgCl$_2$, and 1 mM MnCl$_2$ in PBS pH 6.8) and incubated with biotinylated *Griffonia simplifolica* isolectin B4 (iB4, 1:25, Vector Labs (Burlingame, CA) in Pblec buffer. Antibodies were diluted in 1% BSA, 1% normal donkey serum (NDS), 1% Triton X-100 PBS and muscle samples were incubated in

antibody solution for two days at 4°C. We used the following primary antibodies: FITC-conjugated smooth muscle actin (1:500, Sigma-Aldrich Cat# F3777, RRID:AB_476977), goat anti-endomucin (1:100, R & D Systems (Minneapolis, MN) Cat# AF4666, RRID:AB_2100035) and mouse anti-Neurofilament-M (1:100, Developmental Studies Hybridoma Bank (University of Iowa), Cat# 2H3, RRID:AB_531793). The samples were washed 3 × 20 min each with 0.5% BSA, 0.5% Triton X-100 in PBS at room temperature (RT). Fluorescence detection using secondary antibodies was performed by incubation with Alexa Fluor-488-conjugated streptavidin (1:100, Invitrogen (Carlsbad, CA)) and/or Alexa Fluor-546-conjugated donkey anti-goat or donkey anti-mouse antibodies (1:200, Invitrogen) in 1% BSA, 1% NDS, 1% Triton X-100 in PBS overnight at 4°C. Samples were washed 3 × 20 min each with 0.5% BSA, 0.5% Triton X-100 in PBS and once with PBS at RT and then cleared sequentially (70% and 90% glycerol/PBS, at least 5 hr each) and mounted in 90% glycerol/PBS.

Whole mount retina (*Pitulescu et al., 2010*) and (retinal pigment epithelium (RPE)/choroid/sclera) (*Cashman et al., 2011*) staining was performed as previously described. Samples were stained with biotinylated or Alexa Fluor-488- conjugated iB4 (1:25, Vector Labs), rabbit anti-NG2 (1:200, EMD Millipore (Billerica, MA) Cat# AB5320, RRID:AB_91789), or goat anti-Dll4 (1:100, R & D Systems Cat# MAB1389, RRID:AB_2092985). Fluorescence detection was performed using Alexa Fluor-488-conjugated streptavidin, Alexa Fluor-546 or 633-conjugated secondary antibodies and Alexa Fluor 546-conjugated Phalloidin (Invitrogen). DNA was stained with 1 µM 4,6'-diamidino-2-phenylindole (DAPI) or 10 µg/ml Hoechst (both from Invitrogen) in PBS for 10 min at RT and retinas and (RPE/choroid/sclera) were mounted in FluoromountG (Southern Biotech).

## Muscle histology

Mice were anesthetized (150 mg/kg ketamine and 13 mg/kg xylazine) and treated by intravenous injection of 400 U heparin prior to thoracotomy. The right atrium was severed and the mice were maximally vasodilated by infusing, via the left ventricle, 20 ml normal saline containing 1 g/l adenosine, 4 mg/l papaverine and 100 µg/ml heparin followed by 10 ml 2% formalin. The skin was removed and entire hindlimbs were fixed in 10% formalin (RT, 24 hr). Calf and adductor muscles were dissected *en block* from fixed hindlimbs, dehydrated and embedded in paraffin. Cross sections (7 µm) were prepared and subjected to antigen retrieval using 1x antigen unmasking solution (Vector Labs). The sections were blocked and permeabilized in 10% normal goat serum, 0.1% Triton X-100 in PBS for (RT, 1 hr) and incubated with Alexa Fluor 488-conjugated IsolectinB4 (1:25, Vector Labs) and primary antibodies, mouse anti-smooth muscle actin (1:500, Sigma-Aldrich Cat# A5228, RRID:AB_262054) and rat anti-CD31 (1:50, BD Biosciences Cat# 558736, RRID:AB_397095) in 1% BSA in PBS (RT, 2 hr). Sections were washed 3 × 5 min each with PBS and incubated with Alexa Fluor 546-goat anti mouse and Alexa Fluor 488-goat anti rat antibodies in 1% BSA PBS for 1 hr at RT. Following washing as above, DNA was stained with DAPI, sections mounted in FluoromountG (Southern Biotech) and imaged on a TCS SP2 Leica confocal microscope.

## Microscopy and image analysis

Whole mount muscle and retinal vasculature imaging was done on a Zeiss stereomicroscope or a TCS SP2 Leica confocal microscope. Maximum projection confocal images of the adductor muscle vasculature were generated from z-stacks (30–300 µm, 1–10 µm step size depending on specimen size, staining and objective used) acquired starting at the medial surface of the adductor muscle specimens. To visualize large areas of the vasculature on the confocal microscope, a tile-scanning technique was employed whereby multiple overlapping (20–30% overlap) maximum projection images were acquired with a 10x or 20x objective and a composite image was constructed by arraying the individual images in Photoshop. Quantification of vascularized area in whole mount retinas was done from fluorescence stereomicroscopic images using ZEN software (Zeiss). Retinal angiogenic front vascular density, endothelial sprouts and filopodia were quantitated using ImageJ and maximum projection confocal images acquired with a 10x, 20x and 63x objective respectively. The number of tip cells and filopodia was measured as described (*Pitulescu et al., 2010*).

## Murine lung endothelial cells (MLEC)

Primary MLEC cultures were prepared as described (*Kuhlencordt et al., 2004*) with minor modifications. Briefly, lungs were harvested aseptically, rinsed in Dulbecco's modified eagle medium

(DMEM), cut into small pieces and digested (1 hr, 37°C) in 1.7 mg/ml collagenase (Worthington). Lung digests were triturated by pipetting repeatedly through a 10 ml pipette fitted with a 1 ml pipette tip, passed through a 40 μm filter, and the cells obtained were cultured (2 days) on gelatin-coated plates in MLEC medium (20% fetal bovine serum, 38% DMEM, 38% Ham's F-12 supplemented with 100 μg/mL endothelial cell growth supplement (ECGS, Biomedical Technologies), 4 mM L-glutamine, 100 μg/mL heparin, and 1% penicillin/streptomycin (Life Technologies). Endothelial cells were isolated by selection with rat anti-mouse intercellular adhesion molecule 2 (ICAM2) antibody (BD Biosciences Cat# 553326, RRID:AB_394784) -coupled sheep anti-rat Ig magnetic beads (Invitrogen) and cultured for an additional 3–4 days. The cells were then subjected to a second round of selection using magnetic beads and then cultured for an additional 2 days before studies were performed. The purity of the primary endothelial cell cultures was examined by staining live cells with 1,1'-dioctadecyl - 3,3,3',3'-tetramethyl-indocarbocyanine perchlorate acetylated low-density lipoprotein (Dil-Ac-LDL; BT-209; Alfa Aesar (Ward Hill, MA)) and by staining with a PE-conjugated antibody to Cdh5 (1:50, eBioscience Cat# 12-1441-82, RRID:AB_1907346). The staining was examined by fluorescence microscopy and flow cytometry.

The 5-ethynyl-2'-deoxyuridine (EdU) incorporation assays were performed by incubation of cell cultures with 10 μM EdU (6 hr). The cells were processed for detection of EdU incorporation using the Click-iT EdU Alexa Fluor 488 Imaging Kit according to the manufacturer's instructions (Invitrogen).

Confluent MLEC cultures in DMEM/F12 supplemented with 1% FBS were stimulated with 100 ng/ml VEGF-A or bFGF (Peprotech) or incubated in 1% $O_2$.

Immunofluorescence analysis was done using cells fixed with 4% paraformaldehyde (RT, 15 min). The cells were washed (3 × 10 min each) with PBS and subsequently incubated in permeabilizaton/blocking buffer (10% normal goat serum (NGS) or NDS (depending on the species of secondary antibody used), 0.1% Triton X-100 (1 hr, RT), and then incubated with primary antibodies, including PE-conjugated rat anti-Ki-67 (1:200, eBioscience Cat# 12-5698-82, RRID:AB_11150954), mouse anti-α Tubulin (1:500, Sigma-Aldrich Cat# T5168, RRID:AB_477579), rat anti-Cdh5 (1:50, BD Biosciences Cat# 550548, RRID:AB_2244723) and goat anti-Dll4 (1:100, R & D Systems Cat# MAB1389, RRID: AB_2092985) in 1% BSA, 0.1% Triton X-100 (4°C, 12 hr) and washed (3 × 10 min each) with PBS. Fluorescence detection using secondary antibodies was performed by incubation (RT, 2 hr) with appropriate Alexa Fluor- 488, 546 or 633-conjugated secondary antibodies (1:200, Invitrogen). The cells were then washed (3 × 10 min each) with PBS. DNA was stained with 4,6'-diamidino-2-phenylindole (DAPI) or Hoechst (1 μM, Invitrogen). The cells were mounted in FluoromountG (Southern Biotech) and imaged on a TCS SP2 Leica confocal microscope. Fluorescence was quantitated using ImageJ software.

## Endothelial cell tubulogenesis assays

Primary MLECs (1 × $10^5$ cells) in 0.5% FBS DMEM/F12 were seeded in 8 well chamberslides (BD Biosciences) layered with 300 μl polymerized growth factor reduced matrigel (BD Biosciences) and incubated at 37°C for 8 hr. Tubular networks were imaged using a Zeiss inverted microscope.

## Endothelial cell migration assays

Confluent monolayers of primary MLECs in 96 well plates were simultaneously scratched using a 96-pin wound making tool (WoundMaker, Essen Bioscience), rinsed twice with media and wound closure was monitored by automated live cell imaging on an IncuCyte ZOOM system (Essen Bioscience (Ann Arbor, MI)) using a 10x objective. The area between the edges of the wound in images taken at different time intervals was quantitated using ImageJ.

## RNA isolation

To isolate RNA from tissues, mice were perfusion cleared with PBS via the left ventricle. Hindlimb adductor and calf skeletal muscles were harvested en block, snap frozen in liquid nitrogen, and then pulverized into a powder using a CryoPREP impactor (Covaris (Woburn, MA)). Total RNA was extracted with TRIzol (Life Technologies) and was purified using an RNeasy kit (Qiagen). RNA from cells and other tissues homogenized in RLT buffer (Qiagen) was isolated using the RNeasy kit.

## Quantitative RT-PCR assays

We used purified RNA to prepare cDNA using The High Capacity Reverse Transcription Kit (Life Technologies). The expression of mRNA was examined by quantitative PCR analysis using a Quantstudio PCR system (Life Technologies). TaqMan assays were used to quantitate *Cdh5* (Mm00486938_m1), *Dll4* (Mm00444619_m1), *Emr1* (Mm00802529_m1), *Hes1* (Mm01342805_m1), *Hey1* (Mm00468865_m1), *Lfng* (Mm00456128_m1), *Pecam1* (Mm01242584_m1), *Slc2a1* (Mm00441480_m1) and *Vegfa* (Mm01281449_m1). The relative mRNA expression was normalized by measurement of the amount of 18S RNA in each sample using TaqMan assays (catalog number 4308329; Life Technologies).

## RNA sequencing

RNA was isolated using the RNeasy kit (Qiagen). RNA quality (RIN > 9) was verified using a Bioanalyzer 2100 System (Agilent Technologies). Total RNA (10 µg) from independent MLEC isolations (lungs from 4 mice per isolation) was used for the preparation of each RNA-seq library by following the manufacturer's instructions (Illumina). Three independent libraries were examined for each condition. The cDNA libraries were sequenced by Illumina Hi-Seq with a paired-end 40-bp format. Reads from each sample were aligned to the mouse genome (UCSC genome browser mm10 build) using TopHat2 (*Kim et al., 2013*). The average number of aligned reads per library was > 20,000,000. Endothelial cell gene expression was quantitated as fragments per kilobase of exon model per million mapped fragments (FPKM) using Cufflinks (*Trapnell et al., 2010*). Differentially expressed genes were identified using the Cufflinks tools Cuffmerge and Cuffdiff. Differentially expressed genes were defined as those genes that were expressed (Fragments Per Kilobase of exon per Million fragments mapped [FPKM] > 2); absolute log2-fold change > 0.5; $q \leq 0.05$. Gene ontology was examined by Kyoto Encyclopedia of Genes and Genome (KEGG) pathway analysis (*Kanehisa et al., 2012*) with the Database for Annotation, Visualization and Integrated Discovery (DAVID) (*Huang et al., 2009*).

## Transplantation assays

Bone marrow (BM) was harvested by flushing tibias and femurs from at least three 10–12 week old mice with ice cold PBS. Erythrocytes were disrupted by incubation of BM in ACK lysing buffer (Life Technologies). The BM cells were then resuspended in PBS and passed through a 100 µm filter. Cells were counted and mixtures of test BM cells from the indicated genotypes were prepared by mixing test BM cells expressing the CD45.2 allele with competitor BM cells from B6.SJL-*Ptprc*[a] *Pepc*[b]/*BoyJ* mice expressing the CD45.1 allele at a 20 test:80 competitor cell ratio. $1 \times 10^6$ total BM cells were intravenously injected via the tail vein into lethally irradiated (11 Gy) 10–12 week old CD45.1/CD45.2 heterozygous female mice. Transplanted mice were maintained on antibiotic water for the first two weeks post transplantation. Blood was harvested via the retroorbital sinus using heparinized capillary tubes and EDTA-coated vials at 5 and 20 weeks post transplantation and subjected to flow cytometry analysis.

## Complete blood cell (CBC) analysis

CBC analysis was done using a HemaTrue hematology analyzer (Heska (Loveland, CO)) by the Department of Animal Medicine, University of Massachusetts Medical School.

## Flow cytometry

Blood was washed in PBS, stained with live/dead fixable blue dead cell staining kit (Invitrogen), washed in PBS and blocked in 2% FBS-PBS / 0.02% sodium azide plus Fc-block (Anti-CD16/32 antibody 1:200, BD Biosciences). Surface antigens were detected by incubation for 30 min at 4°C with conjugated antibodies including CD45.1-eFluor 450 (eBioscience Cat# 48-0453-82, RRID:AB_1272189), CD45.2-FITC (BD Biosciences Cat# 553772, RRID:AB_395041), CD3e-APC (BD Biosciences Cat# 561826, RRID:AB_10896663), CD19-APC-H7 (BD Biosciences Cat# 560143, RRID:AB_1645234), CD11b-PE (BD Biosciences Cat# 562287, RRID:AB_11154216) and GR1-Alexa Fluor 700 (BioLegend Cat# 108422, RRID:AB_2137487). Following washing with 2% FBS-PBS 0.02% sodium azide, red cells were lysed and leukocytes fixed by incubating in lyse/fix solution (BD Biosciences). Cells were washed with PBS and analyzed on an LSR-II cytometer (Becton Dickenson). Data were processed using FlowJo Software (Tree Star).

## Immunoblot analysis

Cell extracts were prepared using Triton lysis buffer (20 mM Tris at pH 7.4, 1% Triton X-100, 10% glycerol, 137 mM NaCl, 2 mM EDTA, 25 mM β-glycerophosphate, 1 mM sodium orthovanadate, 1 mM phenylmethylsulfonylfluoride, 10 μg/mL of aprotinin plus leupeptin). Extracts (20–50 μg of protein) were examined by protein immunoblot analysis by probing with antibodies to Cdh5 (BD Biosciences Cat# 550548, RRID:AB_2244723), cJun (Cell Signaling Technology Cat# 9165L, RRID:AB_2129578), pSer[63]-cJun (Cell Signaling Technology (Danvers, MA) Cat# 9261S, RRID:AB_2130162), Dll4 (R&D Systems Cat# MAB1389, RRID:AB_2092985), pERK (Cell Signaling Technology Cat# 4370S, RRID:AB_2281741), ERK (Cell Signaling Technology Cat# 4695P, RRID:AB_10831042), GAPDH (Santa Cruz Biotechnology Cat# sc-365062, RRID:AB_10847862), JNK (BD Biosciences Cat# 554285, RRID:AB_395344), pJNK (Cell Signaling Technology Cat# 4668P, RRID:AB_10831195), cleaved Notch1 (NICD) (Cell Signaling Technology Cat# 4147S, RRID:AB_2153348), and αTubulin (Sigma-Aldrich Cat# T5168, RRID:AB_477579). Immune complexes were detected using the Odyssey infrared imaging system (LI-COR Biotechnology (Lincoln, NE)).

## Statistical analysis

Differences between groups were examined for statistical significance with an unpaired Student's test with equal variance or a log-rank (Mantel-Cox) test for determining significance of Kaplan-Meier survival curves. All studies were performed with at least three biological replicates. The number of biological replicates is stated in the figure legends.

## Acknowledgements

We thank Ralf H. Adams for providing the *Cdh5-Cre/ERT2* mice, Stacey Russell (Musculoskeletal Imaging Core (NIH grant S10RR023540) at the University of Massachusetts Medical School) for assistance with μCT imaging, Sudeshna Fisch and Ronglih Liao (Partners Cardiovascular Physiology Core at Brigham & Women's Hospital) for assistance with echocardiography and myocardial infarction studies, Rajendra Kumar-Singh for assistance with CNV assays, Nathan Lawson for discussions, Myoung Sook Han for providing M$^{WT}$, M$^{2KO}$, Ø$^{Ctrl}$ and Ø$^{KO}$ mice, Xiaoyun Huang, Vicky Benoit and Tamera Barrett for expert technical assistance, and Kathy Gemme for administrative assistance. The RNA-seq data has been deposited in the Gene Expression Omnibus (GEO) database with accession number GSE71159. These studies were supported by grants DK107220 (RJD) and HL09122 (JFK) from the National Institutes of Health. RJD is an investigator of the Howard Hughes Medical Institute. The authors declare no competing financial interests.

# Additional information

### Competing interests

RJD: Reviewing editor, *eLife*. The other authors declare that no competing interests exist.

### Funding

| Funder | Grant reference number | Author |
|---|---|---|
| National Institute of Diabetes and Digestive and Kidney Diseases | R01DK107220 | Roger J Davis |
| National Heart, Lung, and Blood Institute | R01HL09122 | John F Keaney Jr |
| Howard Hughes Medical Institute | Investigatorship | Roger J Davis |

The funders had no role in study design, data collection and interpretation, or the decision to submit the work for publication.

### Author contributions

KR, Conceived and designed these studies, Performed experiments and analyzed data, Drafting or revising article; KS, Performed blood pressure and heart rate measurements, Acquisition of data,

Analysis and interpretation of data; SC, Assisted with vasocontraction/vasorelaxation assays, Acquisition of data, Analysis and interpretation of data; JFK, RJD, Conceived and designed these studies, Analysis and interpretation of data, Drafting or revising the article

## Author ORCIDs
Roger J Davis, http://orcid.org/0000-0002-0130-1652

## Ethics

Animal experimentation: This study was performed in strict accordance with the recommendations in the Guide for the Care and Use of Laboratory Animals of the National Institutes of Health. All of the animals were handled according to approved institutional animal care and use committee (IACUC) protocols of the University of Massachusetts Medical School, Tufts University School of Medicine, and Brigham & Women's Hospital.

## Additional files

### Major datasets

The following dataset was generated:

| Author(s) | Year | Dataset title | Dataset URL | Database, license, and accessibility information |
| --- | --- | --- | --- | --- |
| Ramo K, Sugamura K, Craige S, Keaney JF, Davis RJ | 2016 | Suppression of ischemia in arterial occlusive disease by JNK-promoted native collateral artery development | http://www.ncbi.nlm.nih.gov/geo/query/acc.cgi?acc=GSE71159 | Publicly available at the NCBI Gene Expression Omnibus (accession no: GSE71159) |

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
