## [Decision Letter]

Thank you for submitting your article "Suppression of ischemia in arterial occlusive disease by *JNK*-promoted native collateral artery development" for consideration by *eLife*. Your article has been reviewed by two peer reviewers, and the evaluation has been overseen by Fiona Watt as the Reviewing and Senior Editor. One of the two reviewers has agreed to reveal his identity: Bijan Modarai (Reviewer #2).

The reviewers have discussed the reviews with one another and the Reviewing Editor has drafted this decision to help you prepare a revised submission.

Summary:

This thorough report evaluates the necessity of *JNK* signaling in appropriate collateral circulation formation, both in development and in the setting of adult femoral artery ligation. The authors perform a number of in vivo experiments focusing on the loss of *JNK* (1/2 & 1/2/3) as well as upstream MAP kinases, MLK1 and 2, to determine the necessity of proper compensatory angiogenic responses. While it is initially shown that *JNK* and MLK loss results in improper collateral circulation in the adult (Figure 1), it is later demonstrated to be induced by developmental defects during vascular development (Figure 4), causing the study to pivot to a focus on vessel sprouting in developmental angiogenesis. Using a model of retinal angiogenesis, the authors demonstrate that loss of *JNK* signaling leads to increases in filopodia and tip cells with no differences in pericyte coverage (Figure 6), and follow-up gene expression studies of primary endothelial *JNK* knockouts point to defects in Dll4/Notch signaling as the primary mechanism for alterations in sprouting (Figure 7). The investigation of *JNK* signaling in vascular development is thorough and provides various outstanding technical approaches.

Required revisions:

1) Given common origins of hematopoietic and vascular compartments, some work could be done to determine if vascular defects from lost *JNK* signaling are specific to endogenous mature endothelial cells or changes in endothelial progenitor cells.

2) Some investigation of monocyte/macrophage distribution (in hindlimb muscle and circulation) and phenotype at steady state and after induction of ischaemia in *JNK* deficient mice is required. These cells are key regulators of collateralization.

3) The authors should discuss the translational value of their work. Would they expect to find deficiencies in the MLK-*JNK* pathway in patients with critical limb ischaemia, for example? Could any of the findings of this study inform novel endeavors to promote collateralisation in these patients? Would it be possible to identify patients with peripheral arterial disease that do not benefit from extensive native collaterals and would therefore be at risk of developing critical limb ischaemia?

4) One mouse in the E^3KO^ group had a normal limb after femoral artery ligation (Figure 1). How can the authors explain this in view of the severe phenotype demonstrated in the other mice in that group? Was the ligation carried out correctly in this mouse?

---

## [Author Response]

*Required revisions:*

*1) Given common origins of hematopoietic and vascular compartments, some work could be done to determine if vascular defects from lost JNK signaling are specific to endogenous mature endothelial cells or changes in endothelial progenitor cells.*

The common origins of the hematopoietic and endothelial compartments raise questions concerning the possible role for early hematopoietic progenitor cells in the hind limb ischemia response. To test this possibility, we examined the effect of *Jnk* gene ablation in hematopoietic stem cells using Vav1-cre; this analysis demonstrated that *JNK*-deficiency in hematopoietic stem cells caused no change in recovery from hind limb ischemia (Figure 1—figure supplement 6).

The Cdh5-cre driver we used may cause gene ablation in both endothelial progenitor cells and mature endothelial cells. Consequently, the phenotype we describe could result from *JNK*-deficiency in either of these cell types. To test whether *JNK*-deficiency in endothelial cells is required for the failure to recover from hind limb ischemia, we used an inducible Cdh5ERT2-cre driver to ablate *Jnk* genes in adult mice; this analysis demonstrated that *JNK*-deficiency in endothelial cells caused no change in recovery from hind limb ischemia (Figure 4). This experiment demonstrates that there is no role for *JNK* in endothelial progenitor cells or mature endothelial cells in the response of adult mice to hind limb ischemia. *JNK* in adult mice therefore does not regulate revascularization mediated by either endothelial progenitor cells or mature endothelial cells.

Our analysis demonstrates that the failure to recover from hind limb ischemia caused by *JNK*-deficiency in endothelial cells reflects an embryonic developmental defect in collateral vessel formation. Whether this is caused by a primary defect in embryonic endothelial progenitor cells or a primary defect in embryonic mature endothelial cells is unclear. We note that most studies of endothelial progenitor cells (relevant to revascularization) have been done in adult animals rather than embryos. Moreover, the identity of Cdh5+ Flt4+ VEGFR2+ cells as endothelial progenitors or monocytoid cells that can adopt an endothelial-like phenotype is controversial. Given this background, we believe that it is beyond the scope of the present study to define roles for endothelial progenitor cells and mature endothelial cells during embryonic development of collateral vessels in muscle. However, we have obtained direct evidence for relevant defects caused by *JNK*-deficiency in mature endothelial cells, including reduced Notch signaling (Figure 7 and Figure 7—figure supplement 1) and increased hypersprouting during developmental angiogenesis (Figure 6 and Figure 6—figure supplement 1–Figure 6—figure supplement 3).

These data support the conclusion that *JNK*-deficiency in mature endothelial cells contributes to the embryonic developmental defect.

*2) Some investigation of monocyte/macrophage distribution (in hindlimb muscle and circulation) and phenotype at steady state and after induction of ischaemia in JNK deficient mice is required. These cells are key regulators of collateralization.*

We agree – monocyte/macrophages are implicated in post-occlusion collateral artery remodeling in adult mice. It is for this reason that we examined the effect of *JNK*-deficiency in hematopoietic cells (using Vav1-cre) and myeloid cells (using LysM-cre) in hindlimb ischemia (Figure 1—figure supplement 6); this analysis demonstrated no role for *JNK* in macrophages during hind limb ischemia. We also examined macrophages in endothelial cell specific *JNK* knockout mice that fail to recover after hind limb ischemia. Flow cytometry demonstrated no significant differences in the number of circulating macrophages (Figure 1—figure supplement 2). Moreover, no defect in macrophage recruitment to the calf muscle before and following hind limb ischemia in mice with endothelial cell-specific *JNK*-deficiency (Figure 1—figure supplement 8). It is most likely that the absence of a macrophage phenotype in our studies reflects the embryonic developmental origin of the defect in muscle collateral vessels that we describe (rather than collateral artery remodeling in adult mice).

*3) The authors should discuss the translational value of their work. Would they expect to find deficiencies in the MLK-JNK pathway in patients with critical limb ischaemia, for example? Could any of the findings of this study inform novel endeavors to promote collateralisation in these patients? Would it be possible to identify patients with peripheral arterial disease that do not benefit from extensive native collaterals and would therefore be at risk of developing critical limb ischaemia?*

Thank you – this is an excellent point. Since the mechanism we describe reflects an embryonic developmental defect, it is not clear that our conclusions concerning the role of endothelial cell *JNK* during recovery from ischemia can be directly translated to adults with critical limb ischemia. However, germ-line mutations in *JNK* pathway genes in humans may contribute to increased risk for poor outcomes following exposure to an ischemic insult. The increased use of genomic sequencing in patient care may allow the identification of people with increased risk for ischemic injury. We have revised the manuscript in the Conclusion to note this point.

*4) One mouse in the E^3KO^ group had a normal limb after femoral artery ligation (Figure 1). How can the authors explain this in view of the severe phenotype demonstrated in the other mice in that group? Was the ligation carried out correctly in this mouse?*

Our data policy is to report all observations. No data have been censored from the analysis presented in this manuscript. We do not know why one E^3KO^ mouse exhibited a mild phenotype (no planatar flexion and necrosis of one toe) in this experiment. However, we believe that full disclosure and transparency of data presentation is a critical aspect of the scientific method.